# BREAKING RANK BOTTLENECKS
# IN KNOWLEDGE GRAPH EMBEDDINGS

## ABSTRACT

Many knowledge graph embedding (KGE) models for link prediction use powerful encoders. However, they often rely on a simple hidden vector-matrix multiplication to score subject-relation queries against candidate object entities. When the number of entities is larger than the model's embedding dimension, which is often the case in practice by several orders of magnitude, we have a linear output layer with a *rank bottleneck*. Such bottlenecked layers limit model expressivity. We investigate both theoretically and empirically how rank bottlenecks affect KGEs. We find that, by limiting the set of feasible predictions, rank bottlenecks hurt the ranking accuracy and distribution fidelity of scores. Inspired by the language modelling literature, we propose KGE-MOS, a mixture-based output layer to break rank bottlenecks in many KGEs. Our experiments show that KGE-MOS improves ranking performance of KGE models on large and dense datasets at a low parameter cost.

## 1    INTRODUCTION

Knowledge graph completion (KGC) aims to predict missing triples in a knowledge graph (KG). For example, KGC models can find missing connections in a biomedical KG to reveal potential drug-disease associations (Himmelstein et al., 2017; Breit et al., 2020; Bonner et al., 2022), and are used as recommender systems in social networks or e-commerce (Hu et al., 2020). Most KGC models learn  knowledge graph embeddings (KGEs) for entities and relations to obtain a score indicating what objects are likely for a given subject-relation pair (Nickel et al., 2015; Ji et al., 2022).

De facto KGE models score object entities by simply multiplying an embedding of the subject-relation query with the entity embedding matrix. This is the case for both bilinear or neural network approaches, even when they use powerful encoders like graph neural networks or language models (Ali et al., 2022). This vector-matrix multiplication linearly maps a hidden state of relatively low dimension (typically, $10^2$ to $10^4$) to a high-dimensional output space corresponding to the number of entities (typically, $10^4$ to $10^6$ for medium KGs, and up to $10^9$ for industry-scale KGs (Sullivan, 2020)). Such linear output layers introduce low-rank constraints, often referred to as *rank bottlenecks*. Yang et al. (2017) showed that rank bottlenecks hurt language modelling (LM) perplexity.

Even though the number of KG entities is usually much larger than the vocabulary size in LM, rank bottlenecks are not yet well-explored in KGEs. Previous works have shown ranking expressivity bounds for specific bilinear KGEs (Trouillon et al., 2017; Wang et al., 2017; Balažević et al., 2019), but such bounds are missing for more general KGEs or in the context of other prediction tasks. We show how bottlenecking KGEs impacts their predictions in large and dense graphs. We define dimensional requirements in the context of three tasks for expressing either (i) all possible patterns in graphs of a certain size (*universality*, Figure 2) or (ii) specific patterns in a given graph (*realisability*).

To break the rank bottleneck, we introduce KGE-MOS, which uses a mixture of softmaxes (MoS) (Yang et al., 2017) as a non-linear output layer. We evaluate KGE-MOS on five KGE models across knowledge graphs of increasing size. On the larger and denser datasets, KGE-MOS improves both the ranking accuracy and probabilistic fit of probabilistic predictions at a lower parameter cost than simply increasing the KGE dimension.

**Contributions.** We (i) show the implications of rank bottlenecks with necessary bounds for universality and realisability on three KGC prediction tasks that apply to any bilinear and neural network KGE model (Section 3). In Section 4, we (ii) concretise sufficient dimensions to perfectly break

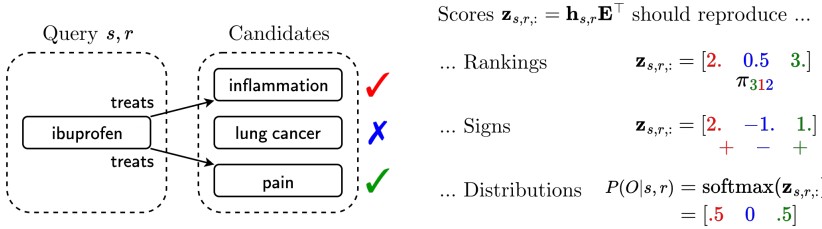

Figure 1: KGE models are used in various prediction tasks, each with their own expressivity needs.

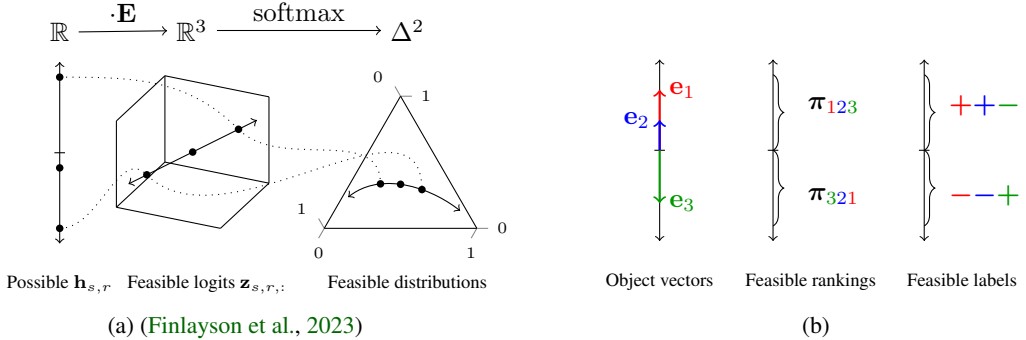

(a) (Finlayson et al., 2023)  (b)

Figure 2: **Rank bottlenecks leave many score / ranking / sign-label reconstructions unfeasible** when $d < |\mathcal{E}| - 1$ (illustrated on a toy KGE, $d = 1$, $|\mathcal{E}| = 3$). (a) Projection of the hidden space of queries $(s, r, ?)$ by the matrix of object vectors $\mathbf{E}$. Logits lie in a 1D linear subspace. Probabilities after softmax lie in a 1D smooth manifold in the probability simplex, leaving all other points unfeasible. (b) Division of the hidden state space by object vectors $\mathbf{e}_o$ into regions corresponding to different rankings / labels. Many configurations (e.g., $\boldsymbol{\pi}_{312}$ or $+-+$) are unfeasible.

bottlenecks in some tasks and show that the task difficulty depends on the graph connectivity. We (iii) introduce KGE-MoS, a simple output layer that breaks rank bottlenecks and improves performance in large-scale benchmarks at a low parameter cost (Section 5).

## 2 BACKGROUND

### 2.1 RANK BOTTLENECKS IN HIGH-DIMENSIONAL OUTPUT LAYERS

Neural networks often map a lower-dimensional hidden representation to a higher-dimensional output space – for example, when they make a prediction among a large set of entities or large vocabularies. When this mapping is linear, as is standard in many architectures, it imposes low-rank constraints that fundamentally limit the kinds of functions the model can represent.

Concretely, consider a hidden vector $\mathbf{h} \in \mathbb{R}^d$ that is projected to an output vector $\mathbf{z} \in \mathbb{R}^m$ by a linear layer $\mathbf{z} = \mathbf{h}\mathbf{W}^\top$ where $\mathbf{W} \in \mathbb{R}^{m \times d}$. Regardless of how expressive the upstream encoder to $\mathbf{h}$ is, the output $\mathbf{z}$ is always confined to the column space of $\mathbf{W}$, which has dimension at most $d$. In other words, the model can only produce outputs lying in a $d$-dimensional linear subspace of the $m$-dimensional output space. When $d \ll m$, this restriction is known as a *rank bottleneck*.

Rank bottlenecks are typically benign in intermediate layers of a neural network because repeated non-linear transformations can "reshape" the linear subspace to meaningful manifolds (e.g., as seen in autoencoders (Hinton & Zemel, 1993)). However, at the final layer, where predictions are consumed directly by, e.g., a softmax or sigmoid layer, this constraint imposes rigid limitations. This phenomenon has been well-studied in language modeling, where the number of output classes (words) exceeds the hidden size by orders of magnitude. Previous work (e.g., (Yang et al., 2017)) showed that

low-rank output layers systematically restrict the model's ability to fit empirical distributions, hurting perplexity and limiting expressivity.

In knowledge graph embedding models (KGE), the situation is even more severe because the output space – which corresponds to all candidate object entities – can range from tens of thousands to millions, typically far larger than common language model vocabularies. Yet, most KGEs have a final output layer that is linear with low-rank constraints, including neural models that attempt to improve the expressivity of the standard bilinear KGE models.

## 2.2 KNOWLEDGE GRAPH COMPLETION

A knowledge graph (KG) is a common data structure for representing relational information. Let $\mathcal{E}$ be a set of entities and $\mathcal{R}$ a set of relations. A KG $\mathcal{G} = \{(s, r, o) \in \mathcal{E} \times \mathcal{R} \times \mathcal{E}\}$ is a set of triples where $s \in \mathcal{E}$ is a subject, $r \in \mathcal{R}$ is a relation, and $o \in \mathcal{E}$ is an object. In the knowledge graph completion (KGC) task, only some triples $\mathcal{G}_{\text{obs}} \subset \mathcal{G}$ are observed, and the task is to predict missing triples $\mathcal{G} \setminus \mathcal{G}_{\text{obs}}$ by predicting suitable objects for $(s, r, ?)$ queries. This is typically cast as a ranking problem where candidate objects are evaluated and ranked; correct answers are objects $o$ such that $(s, r, o) \in \mathcal{G}$ (Nickel et al., 2015).

A knowledge graph embedding model (KGE) is a function $\phi : \mathcal{E} \times \mathcal{R} \times \mathcal{E} \to \mathbb{R}$ that assigns a score $z_{s,r,o}$ to each triple by embedding entities and relations.[1] Here, higher scores correspond to more likely triples. We use $\mathbf{e}_s, \mathbf{e}_o \in \mathbb{R}^d$ to denote the embeddings of subject and object entities respectively and $\mathbf{w}_r \in \mathbb{R}^{d_r}$ to denote the embedding of a relation $r$ (often, $d_r = d$). Let $\boldsymbol{\mathcal{Z}} \in \mathbb{R}^{|\mathcal{E}| \times |\mathcal{R}| \times |\mathcal{E}|}$ be the scores calculated for each triple arranged in a tensor. A KGE answers object prediction queries $(s, r, ?)$ by ordering entities by decreasing values of $\mathbf{z}_{s,r,:} \in \mathbb{R}^{|\mathcal{E}|}$.

### 2.2.1 OBJECT PREDICTION AS A HIDDEN VECTOR-MATRIX MULTIPLICATION

In this work, we focus on KGEs of the form

$$\phi(s, r, o) = \mathbf{h}_{s,r}^\top \mathbf{e}_o. \tag{1}$$

Here $\mathbf{h}_{s,r} \in \mathbb{R}^d$ is a representation of the subject and relation, and is typically the result of an encoding function $h(\mathbf{e}_s, \mathbf{w}_r)$. Most methods use a simple dot product with the object embedding $\mathbf{e}_o$ to compute a score, enabling fast scoring for object prediction. Indeed, let $\mathbf{E} \in \mathbb{R}^{|\mathcal{E}| \times d}$ be the embeddings of all entities stacked in a matrix. The computation of scores for a fixed subject-relation pair $(s, r)$ against all object entities becomes the efficient vector-matrix multiplication

$$\mathbf{z}_{s,r,:} = \mathbf{h}_{s,r} \mathbf{E}^\top. \tag{2}$$

This linear scoring against entity embeddings appears, as their name suggests, in all *bilinear models* such as RESCAL (Nickel et al., 2011), DISTMULT (Yang et al., 2014), COMPLEX (Trouillon et al., 2017), CP (Lacroix et al., 2018) and SIMPLE (Kazemi & Poole, 2018). The same can be derived for TUCKER (Balažević et al., 2019), a model that generalises all the above bilinear models as tensor decompositions. We detail how to rewrite each of these models as $\mathbf{h}_{s,r}^\top \mathbf{e}_o$ in Appendix D.1.

*Neural KGEs* are also of the form in Equation 1. They use a powerful neural encoder to embed subject and relation queries into a hidden state $\mathbf{h}_{s,r} = \text{NeuralNet}(\mathbf{e}_s, \mathbf{w}_r)$. Then they project the hidden state to logits in the object space using a simple linear layer, which recovers $\mathbf{h}_{s,r} \mathbf{E}^\top$. Neural encoders can be CNNs like CONVE (Dettmers et al., 2017; Balažević et al., 2018), graph convolutional networks like COMPGCN (Schlichtkrull et al., 2017; Vashishth et al., 2019), transformers (Galkin et al., 2020) or language models (Wang et al., 2022a; Choi et al., 2021; Wang et al., 2022b; Liu et al., 2022). See Appendix D.1 for details. In Section 7 we discuss additional KGC methods that do not score against the object embedding matrix and are outside the scope of this work.

### 2.2.2 OUTPUT FUNCTIONS AND TASK OBJECTIVES

The introduction of new KGEs went along with the introduction of new training methods and loss functions (Ruffinelli et al., 2020; Ali et al., 2022). For a detailed overview, see Appendix D.2.

---

[1]We use $a$, $\mathbf{a}$, $\mathbf{A}$, $\boldsymbol{\mathcal{A}}$ to denote scalars, vectors, matrices, and tensors respectively. See Appendix A for indexing and slicing notation.

While KGEs are usually evaluated on ranking metrics, the choice of the output and corresponding loss functions often reflects different underlying task objectives. We introduce three such tasks of increasing difficulty.

**Ranking reconstruction (RR)** Given a query of the form $(s, r, ?)$, RR aims to assign scores such that each true target object $o$ receives a higher score $z_{s,r,o}$ than any non-target object $o'$, i.e., $z_{s,r,o} > z_{s,r,o'}$ for all $o$ and $o'$ with $(s, r, o) \in \mathcal{G}$ and $(s, r, o') \notin \mathcal{G}$. RR is typically achieved with margin-based loss functions (Bordes et al., 2013; Yang et al., 2014).

**Sign reconstruction (SR)** For a query $(s, r, ?)$, sign reconstruction aims to learn a scoring function such that $z_{s,r,o} > 0$ if $(s, r, o) \in \mathcal{G}$, and $z_{s,r,o} < 0$ otherwise. This objective can be viewed as binary classification over triples (Trouillon et al., 2017; Dettmers et al., 2017; Pezeshkpour et al., 2020).

**Distributional reconstruction (DR)** Given a query $(s, r, ?)$, distributional reconstruction aims to recover uniform scores $z_{s,r,o} = \tau^+$ for all $(s, r, o) \in \mathcal{G}$ and $z_{s,r,o'} = \tau^-$ for all $(s, r, o') \notin \mathcal{G}$, with $\tau^+$ a high score for true triples and $\tau^- < \tau^+$ a low score for negative triples. For example, $\tau^+ = 1$ and $\tau^- = 0$ corresponds to exact binary reconstruction. This can have a probabilistic interpretations, e.g. when $\tau^- \to -\infty$ for softmax layers, such that any larger $\tau^+$ defines a uniform distribution over the true triples. Distributional reconstruction can be used for downstream reasoning. For example, Arakelyan et al. (2021; 2023) combine triple probabilities to answer complex queries, and Loconte et al. (2023) samples triples, which both require accurate data distributions.

We will say that a KGE is *expressive* for a task under a certain condition (typically, a dimensionality bound on its parameters) if there exists a parameter setting that satisfies the task objective under that condition. Notice that any model expressive for DR or SR is also expressive for RR. Moreover, by setting a decision threshold (e.g., $\frac{\tau^+ + \tau^-}{2}$), we can repurpose a model for DR to a model for SR.

## 3 RANK BOTTLENECKS IN KGES...

In this section, we discuss the limitations of bottlenecked KGEs by showing necessary dimensions for expressing each task objective defined above, and showing how these bounds are not practical for large-scale knowledge graphs. By bottlenecked KGEs, we mean the class of KGE models that use a vector-matrix multiplication for object prediction (see Section 2.2.1) where $d$ is less than the necessary dimension for expressing the task objective.

We define dimensional requirements for expressivity in two contexts:

**Universality** Whether a KGE can represent *all* possible relational patterns for graphs of size $|\mathcal{E}|$.

**Realisability** Whether a KGE can represent patterns consistent with a *particular* graph of size $|\mathcal{E}|$.

To develop our bounds and proofs, we consider the ground-truth adjacency matrix $\mathbf{Y} \in \{0, 1\}^{|\mathcal{E}||\mathcal{R}| \times |\mathcal{E}|}$, where $y_{i,o}$ is 1 if $o$ is an object for the subject-relation pair $i$ and 0 otherwise. Similarly, we consider $\mathbf{Z} = \mathbf{H}\mathbf{E}^\top \in \mathbb{R}^{|\mathcal{E}||\mathcal{R}| \times |\mathcal{E}|}$ where $\mathbf{Z} = [\mathbf{z}_{s,r,:}]_{s,r}$ is a view of the score tensor arranged with subject relation pairs on the first axis and objects on the second, and $\mathbf{H} = [\mathbf{h}_{s,r}]_{s,r}$ is a stacked matrix of the hidden states of subject-relation pairs. Since the rank of $\mathbf{Z}$ is at most $d$, each score vector $\mathbf{z}_{s,r,:}$ is confined to a $d$-dimensional linear subspace of $\mathbb{R}^{|\mathcal{E}|}$. In Figure 2, we show visually how $d < |\mathcal{E}| - 1$ leads to universality bottlenecks in a toy KGE with $d=1$ and $|\mathcal{E}|=3$.

### 3.1 ...FOR SOLVING RANKING RECONSTRUCTION

We first discuss how rank bottlenecks affect a model's ability to correctly order potential objects, as is usual in KGC evaluation protocols. Borrowing the notation of Grivas et al. (2022), let $\boldsymbol{\pi}$ denote a permutation of objects. For example, in Figure 2b, the scores $\mathbf{z}_{s,r,:} = \mathbf{h}_{s,r}\mathbf{E}^\top = [-2 \quad -0.5 \quad 3.]$ would imply that we assign the query $\mathbf{h}_{s,r}$ the ranking $\boldsymbol{\pi}_{321}$ since $z_{s,r,3} > z_{s,r,2} > z_{s,r,1}$. Notice that activations like sigmoid of softmax preserve the ranking.

**Theorem 3.1** (RR universality). *The rank $d$ required for a KGE to represent all possible rankings on graphs with $|\mathcal{E}|$ entities is at least $d \geq |\mathcal{E}| - 1$.*

All proofs are in Appendix B. For example, in Figure 2b (our $d = 1, |\mathcal{E}| = 3$ toy model), we see that only the rankings $\boldsymbol{\pi}_{123}$ and $\boldsymbol{\pi}_{321}$ are feasible. If one query $(s, r, ?)$ has a different ground-truth ranking (e.g., $\boldsymbol{\pi}_{213}$), then even the most powerful encoder for $\mathbf{h}_{s,r}$ cannot achieve the ground-truth ranking. This condition is a lower bound on the universal expressivity conditions of any KGEs such as the ones shown for COMPLEX ($d = |\mathcal{E}||\mathcal{R}|$) (Trouillon et al., 2017) or TUCKER ($d = |\mathcal{E}|$) (Balažević et al., 2019). However, in practice, $d \ll |\mathcal{E}|$ by several orders of magnitude, making strict universality impractical in large-scale KGs.

We show next a requirement for the realisability of a KGE for a particular graph $\mathbf{Y}$. Let $\mathbf{Y}^{\pm} = 2\mathbf{Y} - 1 \in \{-1, +1\}^{|\mathcal{E}||\mathcal{R}| \times |\mathcal{E}|}$ be the sign matrix corresponding to $\mathbf{Y}$ where 0 is mapped to $-1$. In the next result, we use the *sign rank* of $\mathbf{Y}^{\pm}$, which is the minimum rank of any real matrix $\mathbf{M}$ with $\text{sign}(\mathbf{M}) = \mathbf{Y}^{\pm}$. We denote it $\text{srank}(\mathbf{Y}^{\pm})$.

**Theorem 3.2** (RR realisability). *The rank $d$ required for realising rankings that are consistent with a graph $\mathbf{Y}$ is at least $d \geq \text{srank}(\mathbf{Y}^{\pm}) - 1$.*

This result is a lower bound to the conditions for realisability in any particular KGE. For example, Wang et al. (2017) show an upper bound to the rank required for RESCAL or COMPLEX which, as we show in Appendix E.2, is up to $2|\mathcal{R}|$ times larger than the one of Theorem 3.2.

In general, the sign rank of a matrix is much smaller than its rank or size, which explains why ranking reconstruction is easier than distributional reconstruction. Still, it cannot be evaluated in practice because determining the sign rank of a matrix is a NP-hard problem (Alon et al., 1985). The usual approach is to bound the sign rank with other quantities that are easier to compute. In Section 4, we show an upper bound that depends on the KG connectivity and is easy to evaluate in practice.

### 3.2 . . . FOR SOLVING SIGN RECONSTRUCTION

We obtain a similar result for the SR task. Let $\mathbf{y} \in \{+, -\}^{|\mathcal{E}|}$ be the multi-label assignment of a query. For example, in Figure 2b, if we obtain the scores $\mathbf{z}_{s,r,:} = \mathbf{h}_{s,r}\mathbf{E}^{\top} = [-2 \quad -0.5 \quad 3.]$, we assign the query $\mathbf{h}_i$ the multi-label assignment $\mathbf{y} = [- \quad - \quad +]$.

**Theorem 3.3** (SR universality). *The rank $d$ required to represent all possible multi-label assignments on graphs with $|\mathcal{E}|$ entities is at least $d \geq |\mathcal{E}|$.*

For example, in Figure 2b, we see that only the multi-label assignments $++-$ and $--+$ are feasible. For any assignment $\mathbf{y}_j$ that is different from these two, it is impossible to learn a representation $\mathbf{h}_j$ such that the KGE classifies it correctly.

The result for realising sign reconstructions follows directly from the definition of sign rank. We again refer to Section 4 for a useful upper bound on this requirement.

**Theorem 3.4** (SR realisability). *The rank $d$ required for realising the exact multi-label assignments for a graph $\mathbf{Y}$ is at least $d \geq \text{srank}(\mathbf{Y}^{\pm})$.*

### 3.3 … FOR SOLVING DISTRIBUTIONAL RECONSTRUCTION

Standard dimensional considerations show that representing all possible score vectors in $\mathbb{R}^{|\mathcal{E}|}$ requires $d \geq |\mathcal{E}|$ and representing all the probability distributions over $\Delta^{|\mathcal{E}|-1}$ requires $d \geq |\mathcal{E}| - 1$. However, DR only requires representing a restricted family of uniform score vectors with score $\tau^+$ for all $o \in S$ and $\tau^-$ for all $o \notin S$, where $S \subseteq \mathcal{E}$ is a certain support of objects. We confirm next that (i) this restricted family of $\tau^+, \tau^-$ score patterns still spans all of $\mathbb{R}^{|\mathcal{E}|}$, requiring $d \geq |\mathcal{E}|$, (ii) under a relaxed notion of DR where the high/low thresholds $\tau_S^+, \tau_S^-$ can be different for each support, the requirement becomes one less.

**Theorem 3.5** (DR universality). *The rank $d$ required to represent all distributional reconstruction patterns with fixed global thresholds $\tau^+, \tau^-$ on graphs with $|\mathcal{E}|$ entities is at least $d \geq |\mathcal{E}|$. If distributional reconstruction is required only up to an arbitrary offset (i.e., allowing support-specific thresholds), then the requirement becomes $d \geq |\mathcal{E}| - 1$.*

The second notion only requires reconstruction up to an arbitrary offset and meets the usual requirement of softmax $d \geq |\mathcal{E}| - 1$. Notice that applying a softmax function on the scores introduces non-linear interactions but still limits the feasible predictions in a rigid way. Softmax maps the

Table 1: Sufficient embedding dimension $d_{SR}$ for exact sign reconstruction of $(s, r, ?)$ queries on different KGs including inverse relations. **The dimension scales with graph connectivity rather than simply the number of entities.**

| Dataset | #Entities | #Rels | Out-degree | | $d_{SR}^+$ |
|---|---|---|---|---|---|
| | | | Avg | Max | |
| FB15k-237 | 14,541 | 237 | 3.83 | 4,364 | 8,729 |
| Hetionet | 45,158 | 24 | 21.66 | 15,036 | 30,073 |
| ogbl-biokg | 93,773 | 51 | 37.48 | 29,328 | 58,657 |
| openbiolink | 180,992 | 28 | 18.12 | 18,420 | 36,841 |

$d$-dimensional linear subspace of scores to a smooth $d$-dimensional manifold within the probability simplex $\Delta^{|\mathcal{E}|-1}$ (see Figure 2a) (Finlayson et al., 2023). However, the manifold is rigidly constrained to certain shapes. Softmax's log-linearity implies that class probability ratios on the manifold are linear on a logarithmic scale. Ground truths that do not adhere to this constraint cannot be represented. We discuss this phenomenon called *softmax bottleneck* (Yang et al., 2017) in Appendix E.1.

The result for realising distributional reconstructions is the same under both threshold notions.

**Theorem 3.6** (DR realisability). *Realising the exact uniform distributions (with fixed or support-specific thresholds) for a graph $\mathbf{Y}$ requires a model rank $d \geq \mathrm{rank}(\mathbf{Y}) - 1$.*

## 4 A SUFFICIENT BOUND FOR SIGN AND RANKING REALISABILITY

Theorems 3.2 and 3.4 give requirements for realising exact rankings and sign predictions using sign ranks, but sign ranks are NP-hard to compute and cannot be evaluated in practice. In this section, we connect an early result that upper bounds sign ranks by the maximum number of sign changes in any row of a matrix (Alon et al., 1985) and insights from the graph factorisation literature connecting this with node degrees (Chanpuriya et al., 2020), to derive a sufficient condition for realisability expressed in subject-relation pair degrees in knowledge graphs.

**Theorem 4.1.** *Let $\mathbf{Y}^{\pm} \in \{-1, +1\}^{|\mathcal{E}||\mathcal{R}| \times |\mathcal{E}|}$ be the sign matrix representing a KG with subject-relation pairs on the first axis and objects on the second. $\mathbf{Y}^{\pm}$ can be exactly decomposed as $\mathbf{Y}^{\pm} = \mathrm{sign}(\mathbf{H}\mathbf{E}^{\top})$ where $\mathrm{sign}$ is applied element-wise, $\mathbf{H} \in \mathbb{R}^{|\mathcal{E}||\mathcal{R}| \times (2c+1)}$, $\mathbf{E} \in \mathbb{R}^{|\mathcal{E}| \times (2c+1)}$, and $c$ is the maximum out-degree across all subject-relation pairs in the KG.*

The proof is detailed in Appendix B. By definition of the sign rank, this gives the upper bound $\mathrm{srank}(\mathbf{Y}^{\pm}) \leq 2c + 1$ which lets us state the following corollary.

**Corollary 4.2** (Sufficient rank for realising RR and SR). *For any KG with maximum out-degree $c$ across subject-relation pairs, there exists a KGE with rank $d_{SR} = 2c + 1$ that realises perfect rankings and signs for queries $(s, r, ?)$.*

Table 1 presents the theoretical sufficient dimension $d_{SR} = 2c + 1$ for several KGs. In summary, more densely connected datasets are harder to fit in theory. This explains why a dataset like ogbl-biokg, despite having fewer nodes than openbiolink, is more challenging. We remark that the technique used to prove Theorem 4.1 depends on an ordering of the entities, and the value $2c + 1$ is that of the *worst-case ordering*. In practice, the bound can be made tighter using better orderings (see Fig. 4 in Appendix), but finding the optimal ordering is a hard problem and is dataset-specific.

Note that two practical questions remain. Firstly, can existing KGEs, with their specific scoring functions, actually represent the factorisation technique described in Theorem 4.1? With large enough encoding layers, neural KGEs should be able to express this factorization by virtue of neural networks being universal approximators (the $\mathbf{h}_{s,r}$ and $\mathbf{e}_o$ representations are not rigidly tied, which allows to represent the proper embeddings for the factorization). However, bilinear KGEs are more limited. The best known sufficient bound for RESCAL or COMPLEX, shown by Wang et al. (2017), is up to $2|\mathcal{R}|$ times larger than the one of $\mathrm{srank}(\mathbf{Y}^{\pm})$ (see Appendix E.2). It is not clear where the necessary *and* sufficient bound lies between these two. DISTMULT, on the other hand, cannot realise the solution at all due to its symmetric score function. Secondly, even if some KGEs can technically express this solution, can it be efficiently learned through gradient-based optimisation in practice? Interpolation solutions as pictured in Figure 4 are usually in very tight subspaces of the parameter space and hard to converge to in practice. Despite these challenges, this theoretical result provides a valuable guide for dimensionality choices and relating different dataset connectivities.

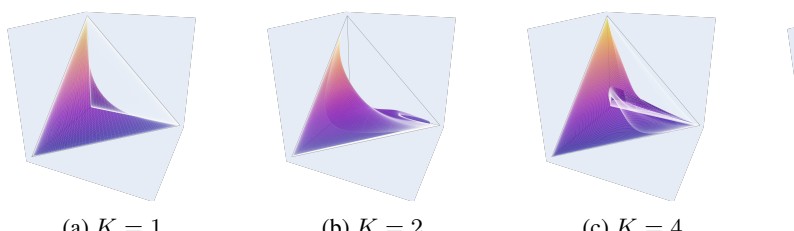

|  |  |  |  |
|:-:|:-:|:-:|:-:|
| (a) $K = 1$ | (b) $K = 2$ | (c) $K = 4$ | (d) $K = 8$ |

Figure 3: Feasible distributions of randomly initialised KGE-MoS output layers in a toy model ($d=2, |\mathcal{E}|=4$). The points are calculated for different inputs $\mathbf{h}_{s,r}$ while $\mathbf{E}$, $\boldsymbol{\omega}_k$ and $f_{\boldsymbol{\theta}_k}$ are fixed. **A higher number of softmaxes $K$ has a higher representational power and can better fit arbitrary ground-truth $P^*(O|s,r)$.**

## 5   BREAKING RANK BOTTLENECKS WITH A MIXTURE OF SOFTMAXES

We now propose a solution to break rank bottlenecks in KGEs. A simple approach is to use a larger embedding dimension $d$ to reflect the graph size or, according to the bound of Theorem 4, its connectivity. However, this quickly becomes impractical for large graphs. Instead, we adapt the *Mixture of Softmaxes* (MoS) layer proposed in language modelling (Yang et al., 2017) to KGEs.

Let $O$ be a random variable over the objects. KGE models use the softmax for object prediction $(s, r, ?)$ to model a categorical distribution $P(O|s,r) = \text{softmax}(\mathbf{h}_{s,r}\mathbf{E}^\top) \in \Delta^{|\mathcal{E}|-1}$, where $\Delta^{|\mathcal{E}|-1}$ is the probability simplex over the objects. This distribution should align with the data distribution (labels of the corresponding triples, normalised to sum to 1). Although softmax is traditionally used for predicting a single correct class, it is often repurposed for multi-label scenarios by retrieving top-$k$ entries, for ranking likely completions (RR) and for modelling generative distributions (DR). Extensive benchmarking efforts have shown that training KGEs with softmax and CE loss leads to the best results (Ruffinelli et al., 2020; Ali et al., 2022).

As shown in Sections 3.3 and 3.1, a bottlenecked KGE model with a softmax activation cannot represent all possible distributions and rankings over the objects. We propose using a KGE-MoS layer, which is a mixture of $K$ softmaxes, to break these rank bottlenecks. Specifically,

$$P(O|s,r) = \sum_{k=1}^{K} \pi_k(\mathbf{h}_{s,r}) \text{softmax}(f_k(\mathbf{h}_{s,r})\mathbf{E}^\top), \quad \sum_{k=1}^{K} \pi_k(\mathbf{h}_{s,r}) = 1, \tag{3}$$

where $\pi_k(\mathbf{h}_{s,r}) \in [0, 1]$ is the prior or mixture weight of the $k$-th component and $f_k(\mathbf{h}_{s,r}) \in \mathbb{R}^d$ is the $k$-th context vector associated with the input $(s, r)$. The prior is parameterised as $\pi_k(\mathbf{h}_{s,r}) = \exp \mathbf{h}_{s,r}^\top \boldsymbol{\omega}_k / \sum_{k'} \exp \mathbf{h}_{s,r}^\top \boldsymbol{\omega}_{k'}$ with $\boldsymbol{\omega}_k \in \mathbb{R}^d$. The context vector $f_k(\mathbf{h}_{s,r})$ is obtained by mapping the hidden state with a component-specific projection with parameters $\boldsymbol{\theta}_k$. In this work, we model each $f_k$ with a two-layer MLP. Notice that only the parameters $\boldsymbol{\theta}_k$ and $\boldsymbol{\omega}_k$ are component-specific, resulting in a low additional parameter cost. $\mathbf{E} \in \mathbb{R}^{|\mathcal{E}| \times d}$ is shared.

The weighted combination of softmaxes is a non-linear transformation of logits which, while not increasing the dimension of the manifold spanned by the KGE, allows for a more complex set of distributions to be represented (see Figure 3). A more algebraic perspective considers the empirical log-probability matrix of the KGE predictions (see Appendix E.1), which resides in a linear subspace with $K = 1$ softmax and in a non-linear subspace when $K > 1$ (Yang et al., 2017). In Appendix F.4.2, we confirm that KGE-MoS breaks the rank bottleneck by measuring the rank of this log-probability matrix for $K = 1$ softmax and $K > 1$ softmaxes on actual KGs.

KGE-MoS is a drop-in solution that can be applied to any KGE method presented in Section 2.2.1 (e.g., CONVE). Simply replacing their output layer results in a high-rank variant (e.g., CONVE-MoS) for a low parameter cost $O(Kd + K|\boldsymbol{\theta}_k|)$, which in our case gives $O(Kd^2)$. In contrast, increasing the dimension of the KGE by a factor $d_{\text{inc}}$ has a cost $O(d_{\text{inc}}(|\mathcal{E}| + |\mathcal{R}|))$ which is usually less scalable as $|\mathcal{E}| \gg Kd$. Notice that while bilinear models (e.g., DISTMULT) can perform subject prediction $(?, r, o)$ scalably as a vector-matrix multiplication, their augmentations (e.g., DISTMULT-MoS) cannot, and must rely on inverse relations $(o, r^{-1}, ?)$. We discuss this in Appendix C.

# 6 EXPERIMENTS

We aim to answer the following research questions: **(RQ1)** "How does the performance of models with the proposed KGE-MOS output layer compare to bottlenecked KGE models?" **(RQ2)** "How does increasing the embedding size, as a simple alternative, compare to KGE-MOS in performance?"

**Datasets**  We evaluate the performance of KGE models on the following standard datasets for link prediction: `FB15k-237` (Toutanova & Chen, 2015), `Hetionet` (Himmelstein et al., 2017), `ogbl-biokg` (Hu et al., 2020), and `openbiolink` (Breit et al., 2020). `FB15k-237` is a commonly used KGC benchmark, but its size is relatively small compared to modern KGs. We choose the other datasets for their larger sizes, as we expect rank bottlenecks to be more impactful in larger graphs due to our theoretical results. See Appendix F.1 for details and statistics.

**Models**  For the baseline KGE models, we use DISTMULT (Yang et al., 2014), COMPLEX (Trouillon et al., 2017), RESCAL (Nickel et al., 2011) and CONVE (Dettmers et al., 2017), each trained using a softmax output layer and inverse relations for subject prediction, as they are standard choices for link prediction (Ruffinelli et al., 2020; Ali et al., 2022). We also include COMPGCN (Vashishth et al., 2019) to study a more expressive neural encoder. We use the best-performing hyperparameters found by Ruffinelli et al. (2020) (see Appendix F.3). We compare the baselines with their high-rank variants DISTMULT-MOS, RESCAL-MOS, etc., where we replace the softmax output layer with the KGE-MOS output layer using $K = 4$ softmaxes. We train each model with bottlenecks at $d = 200$ and $d = 1000$, except for COMPLEX where we halve the dimensions for comparability as the model has a bottleneck of $2d$ (see Appendix D.1). For COMPGCN we only evaluate in the low-dimensional regime due to computational constraints. The hyperparameters for the KGE-MOS output layer are found with a random search on `ogbl-biokg` and detailed in Appendix F.3.

**Metrics**  As usual (Nickel et al., 2015; Ruffinelli et al., 2020; Ali et al., 2022), we assess the models for predicting object queries $(s, r, ?)$ and subject queries $(?, r, o)$ using filtered mean reciprocal rank (MRR) (see Appendix F.2). We measure the distributional fidelity of the models using the negative log likelihood (NLL) of the model predictions on the test set. Whereas previous work (Loconte et al., 2023) directly reports the NLL, we introduce a *filtered NLL* metric which more accurately reflects the predictive performance of a model for KGC. This prevents penalising models that assign low total probability mass outside training triples. See Appendix F.2 for more details.

## 6.1 RESULTS

Table 2 reports our results. Each model is run three times per dataset for statistical significance.

**(RQ1)**  **On the three larger and more densely connected datasets, the best performing models are consistently KGE-MOS models in both low and high dimensional regimes**. In other words, gains are strongest exactly where theory predicts they matter. The best performing model are often DISTMULT-MOS or COMPLEX-MOS, hinting that a simple encoder for the subject-relation with an elaborate object output layer might be the most effective approach. In contrast, on the smallest dataset `FB15k-237`, KGE-MOS does not improve or sometimes even hurts the performance of bottlenecked KGE models. In fact, the best KGE at $d = 200$ performs almost equally to the best KGE at $d = 1000$, hinting that rank bottlenecks are not critical in small-scale datasets. We confirm this in Appendix F.4, where we report similar results on `WN18RR`, a dataset with even lower connectivity than `FB15k-237`. Performance degradation is likely due to overfitting and not instability, as standard deviations on large, dense graphs (Appendix F.4) show minimal deviations across runs.

**(RQ2)**  Comparing models across dimensions, we find that, **on the three larger datasets, KGE-MOS at $d = 200$ obtains almost as strong performance as the bottlenecked baselines at $d = 1000$.** Still, the performance increase of KGE-MOS models is larger at $d = 1000$ than at $d = 200$, suggesting that embedding dimension remains crucial for input representation power. In an additional experiment in Appendix F.4.4, we tried to evaluate bottlenecked KGEs in larger datasets at $d = 5000$, to push the KGE dimension until bottlenecks do not matter. We find that **simply increasing embedding size past some high value requires compromising the training batch size and hurts performance.** This contrasts with the efficieny of KGE-MOS and its low parameter cost.

Table 2: **KGE-MoS improves performance and probabilistic fit of KGEs on the three larger scale datasets**, exactly where theory predicts bottlenecks matter. Average NLL ↓ and MRR ↑. Standard deviations and Hits@ metrics are reported in Appendix F.4. Best results per dataset and dimension in **bold**. We also report the average improvement in NLL and MRR (mixture vs regular) per dataset and dimension. ⋆ indicates statistical significance at $p < 0.05$ in a Wilcoxon signed-rank test pairing models with their mixture variants.

| MODEL | FB15K-237 | | | HETIONET | | | OGBL-BIOKG | | | OPENBIOLINK | | |
|---|---|---|---|---|---|---|---|---|---|---|---|---|
| | NLL | MRR | PARAM | NLL | MRR | PARAM | NLL | MRR | PARAM | NLL | MRR | PARAM |
| *d = 200* | | | | | | | | | | | | |
| DISTMULT | 4.74 | .304 | 3.0M | 6.10 | .250 | 9.0M | 4.83 | .792 | 18.8M | 5.14 | .302 | 36.2M |
| DISTMULT-MOS | 4.65 | .306 | 3.3M | **5.83** | **.277** | 9.4M | 4.65 | .792 | 19.1M | **5.03** | .314 | 36.5M |
| COMPLEX [†] | 4.74 | .303 | 3.0M | 6.10 | .249 | 9.0M | 4.83 | .792 | 18.8M | 5.13 | .301 | 36.2M |
| COMPLEX-MOS [†] | 4.71 | .301 | 3.3M | 5.85 | .269 | 9.4M | **4.65** | **.793** | 19.1M | 5.06 | .313 | 36.5M |
| RESCAL | 4.79 | .258 | 21.9M | 6.13 | .219 | 10.9M | 4.89 | .763 | 22.8M | 5.16 | .303 | 38.4M |
| RESCAL-MOS | 4.65 | .318 | 22.2M | 5.87 | .274 | 11.3M | 4.70 | .780 | 23.2M | 5.04 | **.323** | 38.8M |
| CONVE | **4.48** | **.321** | 5.1M | 6.03 | .252 | 11.1M | 4.94 | .782 | 20.8M | 5.28 | .286 | 38.3M |
| CONVE-MOS | 4.57 | .311 | 5.4M | 5.92 | .263 | 11.4M | 4.77 | .768 | 21.2M | 5.10 | .304 | 38.6M |
| COMPGCN [‡] | 4.80 | .300 | 1.6M | 5.95 | .260 | 4.7M | 5.11 | .749 | 9.5M | 5.43 | .263 | 18.3M |
| COMPGCN-MOS [‡] | 4.86 | .310 | 2.0M | 5.86 | .266 | 5.0M | 4.96 | .744 | 9.6M | 5.43 | .274 | 18.3M |
| *avg -MoS delta* | -.02 | .006 | | ⋆-.19 | ⋆.022 | | ⋆-.17 | .000 | | ⋆-.12 | ⋆.015 | |
| *d = 1000* | | | | | | | | | | | | |
| DISTMULT | **4.56** | **.331** | 15.0M | 6.04 | .288 | 45.2M | 4.89 | .801 | 93.9M | 5.17 | .316 | 181.0M |
| DISTMULT-MOS | 4.72 | .311 | 23.0M | 5.76 | .312 | 53.2M | **4.34** | **.837** | 101.9M | **4.89** | **.347** | 189.0M |
| COMPLEX [†] | 4.71 | .317 | 15.0M | 5.99 | .292 | 45.2M | 4.86 | .806 | 93.9M | 5.12 | .322 | 181.0M |
| COMPLEX-MOS [†] | 4.64 | .314 | 23.0M | 5.78 | .303 | 53.2M | 4.39 | .836 | 101.9M | 4.87 | .345 | 189.0M |
| RESCAL | 4.64 | .307 | 488.5M | 5.93 | .243 | 93.1M | 4.74 | .799 | 195.8M | 5.03 | .328 | 237.0M |
| RESCAL-MOS | 4.63 | .325 | 496.5M | 5.87 | .300 | 101.2M | 4.42 | .824 | 203.8M | 5.00 | .328 | 245.0M |
| CONVE | 4.65 | .301 | 70.1M | 6.06 | .262 | 100.3M | 4.93 | .807 | 149.0M | 5.24 | .308 | 236.2M |
| CONVE-MOS | 4.69 | .316 | 78.2M | **5.71** | **.313** | 108.3M | 4.43 | .817 | 157.0M | 4.91 | .336 | 244.2M |
| *avg -MoS delta* | .02 | .002 | | ⋆-.23 | ⋆.035 | | ⋆-.43 | ⋆.024 | | ⋆-.22 | ⋆.021 | |

[†] Results for COMPLEX use halved $d = 100$ and $d = 500$. It has real and imaginary parameters, leading to a rank bottleneck of $2d$.
[‡] Results for COMPGCN use $d = 100$ on ogbl-biokg and $d = 50$ on openbiolink due to computational constraints.

Table 3: MRR and NLL on ogbl-biokg at $d = 1000$ with different numbers of softmaxes.

| MODEL | #SOFTMAX | NLL ↓ | MRR ↑ | MODEL | #SOFTMAX | NLL ↓ | MRR ↑ |
|---|---|---|---|---|---|---|---|
| DISTMULT | 1 | 4.89 | .801 | COMPLEX | 1 | 4.86 | .806 |
| DISTMULT-MOS | 1 | 4.42 | .821 | COMPLEX-MOS | 1 | 4.46 | .820 |
| DISTMULT-MOS | 2 | 4.37 | .831 | COMPLEX-MOS | 2 | 4.41 | .827 |
| DISTMULT-MOS | 4 | 4.34 | .837 | COMPLEX-MOS | 4 | 4.39 | .836 |
| DISTMULT-MOS | 8 | **4.33** | **.841** | COMPLEX-MOS | 8 | **4.37** | **.838** |

**Mixture ablation** Next, we run an ablation on the number of softmaxes used in KGE-MoS. We use ogbl-biokg as Section 4 suggests it to be the most challenging dataset for bottlenecks due to its high connectivity. We analyse DISTMULT-MOS and COMPLEX-MOS, the best performing models. Table 3 details the results, with each value averaged over three runs (see Table 14 in Appendix for standard deviations). Increasing the number of softmaxes consistently leads to better performance. To confirm that the improvements come from the mixture and not merely the additional encoding power of the projection $f_{\theta_k}$, we also measure the MoS models at $K = 1$ which use $f_{\theta_k}$ without mixing. These models outperform the original models while still bottlenecked, but not as much as the ones breaking the bottleneck with $K > 1$.

**Computational cost** While KGE-MoS has a marginal parameter cost, its computational cost is higher than that of a regular softmax layer. On openbiolink, the dataset with the largest output layer, we find that KGE-MoS at $K = 4$ is between 1.69 and 2.75 times slower than its bottlenecked baseline for a training step. The largest difference is recorded for DISTMULT and DISTMULT-MOS, since DISTMULT is a very simple model. Still, we notice that the number of softmaxes has a negligible impact on inference time. See Table 7 for details. We find most of the overhead is due to the computation of the projections $f_{\theta_k}(\mathbf{h}_{s,r})$. Therefore, if inference time is a concern, this can be mitigated by using a more efficient projection layer.

## 7 RELATED WORK

**Rank bottlenecks**   Rank bottlenecks in output layers of neural networks were first studied in *softmax models* for language modelling (Yang et al., 2017). If the task is to predict a single best completion for a query, low-rank constraints are less problematic (Grivas et al., 2022). However, to predict a distribution over all plausible completions, rank bottlenecks severely limit the expressivity of the model. Yang et al. (2017) proposed a mixture of softmaxes as an output layer to break the bottleneck in language modelling. Yang et al. (2019) proposed a variant, more scalable mixture model, whereas Kanai et al. (2018); Ganea et al. (2019) proposed alternative non-linearities. Grivas et al. (2023) explored rank bottlenecks in clinical and image *multi-label classification* and proposed a discrete Fourier transform output layer to guarantee a minimum number of label combinations to be feasible.

**Expressivity of linear KGEs**   Earlier works (Trouillon et al., 2017; Wang et al., 2017; Kazemi & Poole, 2018; Balažević et al., 2019) have given theoretical guarantees on the expressivity of specific bilinear KGEs for ranking reconstruction. However, the provided conditions (e.g., $d \geq |\mathcal{E}|$) are unrealistic for large datasets, and our work is the first to analyze this issue empirically in specifically large and dense graphs. To the best of our knowledge, we are also the first to provide bounds that apply to any bilinear and neural network KGEs. We also explore expressivity in other prediction tasks previously not considered in expressivity discussions, in light of recent work (Arakelyan et al., 2021; 2023; Harzli et al., 2023; Gregucci et al., 2024) which has also emphasised the importance and lack of well-distributed output scores of KGE models for downstream reasoning. KGE-MOS is reminiscent of ensemble-based KGEs (Wang et al., 2017). However, ensembles (i) are still bottlenecked as they combine models linearly, and (ii) have a high parameter cost (see Appendix G for more details).

**Object prediction without vector-matrix multiplication**   Next, we discuss alternative approaches to KGC that do not use a score function as described in Section 2.2.1 and are outside the scope of this work. *Translation- and rotation-based KGEs* (Bordes et al., 2013; Wang et al., 2014; Lin et al., 2015; Sun et al., 2019) score objects using distance metrics like $\|\mathbf{h}_{s,r} - \mathbf{e}_o\|$, where relations act as geometric transformations in vector space. Translation models are interpretable and fast but are generally not fully expressive (Kazemi & Poole, 2018; Abboud et al., 2020). *Region-based KGEs* (Abboud et al., 2020; Pavlović & Sallinger, 2022) replace vector translations with geometric containment. Entities are embedded as points and relations as boxes. Scoring is based on whether objects lie within the box defined by the subject and relation. These KGEs prioritise spatial interpretability and complex inference patterns (e.g., hierarchical rule injection). They can be shown to be fully expressive, but with large dimension bounds – e.g., $d \geq |\mathcal{E}||\mathcal{R}|$ (Pavlović & Sallinger, 2022) – which are impractical for large datasets. Their performance in a low-dimensional regime should be studied further to investigate other forms of bottlenecks. Finally, some models for KGC do not use a traditional KGE-based scoring function, but have more complex reasoning mechanisms like *graph sampling* (Bi et al., 2022) or *path-based reasoning* (Das et al., 2017; Zhu et al., 2021), which are not subject to our analysis. However, some of these models internally use regular KGE scoring functions (Zhu et al., 2021), so our results could partially be extended to them.

## 8 CONCLUSIONS

In this paper, we showed that rank bottlenecks are a fundamental limitation of many standard KGE models. We demonstrated how these bottlenecks limit the model's expressivity, affecting ranking accuracy and probabilistic fidelity. To address this, we introduced KGE-MOS, a mixture-based output layer that breaks rank bottlenecks efficiently. Our experiments show that KGE-MOS improves the performance of KGEs when working with large KGs, at a low parameter cost. Our findings suggest that exploring high-rank solutions is a promising avenue for advancing KGEs.

**Limitations**   Our sufficient dimension bound for sign and ranking reconstruction highlights that the task difficulty depends on the graph connectivity. More work is needed to tighten the bound and understand whether KGEs can converge to these solutions in practice. Secondly, while KGE-MOS improves performance on larger datasets, the computational cost during inference is higher than for standard KGEs. Using more efficient projection layers can mitigate this. Finally, we did not evaluate KGE-MOS on knowledge graphs with more than millions of entities. Exploring its performance and scalability in settings with significantly larger KGs remains an important avenue for future work.

## REPRODUCIBILITY STATEMENT

We provide information to reproduce our main results (hyperparameter, hardware, initialisation and training details) in Appendix F.3. The paper is submitted with self-contained code for all baselines as well as scripts and instructions to acquire data (which is also explained in Appendix F.1). We provide detailed evaluation metrics in Appendix F.2 to reproduce the main results.

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

## A    TENSOR NOTATION

We borrow the notation from Cichocki et al. (2015) for tensors and matrices, summarised in Table 4.

Table 4: The tensor notation used throughout the paper.

| Notation | Description |
|---|---|
| $\mathcal{A}, \mathbf{A}, \mathbf{a}, a$ | tensor, matrix, vector, scalar |
| $\mathbf{A} = \begin{bmatrix} \mathbf{a}_1 & \ldots & \mathbf{a}_R \end{bmatrix}$ | matrix $\mathbf{A}$ with columns $\mathbf{a}_r$ |
| $\mathbf{A} = \begin{bmatrix} \mathbf{a}_1 \\ \vdots \\ \mathbf{a}_R \end{bmatrix}$ | matrix $\mathbf{A}$ with rows $\mathbf{a}_r$ |
| $a_{i_1, i_2, i_3, \ldots, i_N}$ | scalar of tensor $\mathcal{A}$ obtained by fixing all indices |
| $\mathbf{a}_{:, i_2, i_3, \ldots, i_N}$ | fiber of tensor $\mathcal{A}$ obtained by fixing all but one index |
| $\mathbf{A}_{:, :, i_3, \ldots, i_N}$ | matrix slice of tensor $\mathcal{A}$ obtained by fixing all but two indices |
| $\mathcal{A}_{:, :, :, i_4, \ldots, i_N}$ | tensor slice of $\mathcal{A}$ obtained by fixing some indices |
| $\mathbf{A}_{(n)} \in \mathbb{R}^{I_n \times I_1 I_2 \ldots I_{n-1} I_{n+1} \ldots I_N}$ | mode-$n$ unfolding of $\mathcal{A}$ |

## B    PROOFS

**Theorem 3.1** (RR universality). *The rank $d$ required for a KGE to represent all possible rankings on graphs with $|\mathcal{E}|$ entities is at least $d \geq |\mathcal{E}| - 1$.*

*Proof.* We use a result from Cover (1967); Good & Tideman (1977); Grivas et al. (2022) which states that a linear (affine) classifier over $N$ classes parameterised by a weight matrix $\mathbf{W} \in \mathbb{R}^{N \times d}$ (and bias vector $\mathbf{b} \in \mathbb{R}^N$) followed by any monotonic activation, can only predict a subset of class rankings if $d < N - 1$. $N - 1$ suffices because, in the context of ranking, a constant shift to all scores in the direction of $\mathbf{1}$ does not change the ranking. We are therefore operating in the quotient space $\mathbb{R}^N / \text{span}(\mathbf{1})$ which has dimension $N - 1$. Replacing $\mathbf{W} \in \mathbb{R}^{N \times d}$ with $\mathbf{E} \in \mathbb{R}^{|\mathcal{E}| \times d}$ yields the theorem.

Note that, for object vectors in general position, if we fix $d$, changing $\mathbf{E}$ changes the set of rankings that are feasible, but not the cardinality of the set. The cardinality can be calculated using Stirling numbers with a proof that stems from finding the maximum and generic number of distance-based orderings of $N$ points in a $d$-dimensional space (Cover, 1967; Good & Tideman, 1977; Smith, 2014). The narrower the low-rank $d$, the more class permutations $\boldsymbol{\pi}$ become infeasible. $\qquad \square$

**Theorem 3.2** (RR realisability). *The rank $d$ required for realising rankings that are consistent with a graph $\mathbf{Y}$ is at least $d \geq \text{srank}(\mathbf{Y}^{\pm}) - 1$.*

*Proof.* Let $\mathbf{Z}^{\star}$ be a matrix of model scores that achieves a perfect ranking for every subject-relation pair. For each subject-relation row $i$, there is a threshold $\tau_i \in \mathbb{R}$ such that all true objects have a score strictly greater than $\tau_i$. In other words, $\text{sign}(\mathbf{z}_{i,:}^{\star} - \tau_i \mathbf{1}) = \mathbf{y}_{i,:}^{\pm}$ where $\mathbf{1}$ is the all-ones row vector. Stacking the row thresholds into a matrix $\mathbf{T} = \begin{bmatrix} \tau_1 \\ \vdots \\ \tau_{|\mathcal{E}||\mathcal{R}|} \end{bmatrix} \mathbf{1}^{\top}$, we get $\text{sign}(\mathbf{Z}^{\star} - \mathbf{T}) = \mathbf{Y}^{\pm}$. Since $\mathbf{Z}^{\star}$ has rank at most $d$ and $\mathbf{T}$ has rank 1, $\mathbf{Z}^{\star} - \mathbf{T}$ has rank at most $d + 1$. However, by definition of the sign rank, any matrix $\mathbf{M}$ with $\text{sign}(\mathbf{M}) = \mathbf{Y}^{\pm}$ satisfies $\text{rank}(\mathbf{M}) \geq \text{srank}(\mathbf{Y}^{\pm})$. Therefore, the result is only possible if $d + 1 \geq \text{srank}(\mathbf{Y}^{\pm})$. $\qquad \square$

**Theorem 3.3** (SR universality). *The rank $d$ required to represent all possible multi-label assignments on graphs with $|\mathcal{E}|$ entities is at least $d \geq |\mathcal{E}|$.*

*Proof.* We use the result from Cover (1965); Grivas et al. (2023) which states that a linear (affine) classifier over $N$ classes parameterised by a low-rank weight matrix $\mathbf{W} \in \mathbb{R}^{N \times d}$ (and bias vector $\mathbf{b} \in \mathbb{R}^N$) can only predict a subset of multi-label assignments configurations $\mathbf{y}$ if $d < N$ using the decision rule $\text{sign}(\mathbf{W}\mathbf{x} + \mathbf{b})$ for input $\mathbf{x} \in \mathbb{R}^d$. Replacing $\mathbf{W} \in \mathbb{R}^{N \times d}$ with $\mathbf{E} \in \mathbb{R}^{|\mathcal{E}| \times d}$ yields the theorem. For a linear layer, the number of feasible assignments is upper-bounded by

$2\sum_{i=0}^{d-1}\binom{N-1}{i}$. For an affine layer, more assignments are feasible, but the increase in the number of feasible assignments is less than what would be achieved by increasing the embedding dimension $d$ by one. $\qquad\square$

**Theorem 3.4** (SR realisability). *The rank $d$ required for realising the exact multi-label assignments for a graph $\mathbf{Y}$ is at least $d \geq \mathrm{srank}(\mathbf{Y}^{\pm})$.*

*Proof.* Directly follows from the definition of the sign rank. $\qquad\square$

**Theorem 3.5** (DR universality). *The rank $d$ required to represent all distributional reconstruction patterns with fixed global thresholds $\tau^+, \tau^-$ on graphs with $|\mathcal{E}|$ entities is at least $d \geq |\mathcal{E}|$. If distributional reconstruction is required only up to an arbitrary offset (i.e., allowing support-specific thresholds), then the requirement becomes $d \geq |\mathcal{E}| - 1$.*

*Proof.* **Fixed thresholds.** We show that the span of the score vectors to represent is the span of the standard basis vectors of $\mathbb{R}^n$

Let $|\mathcal{E}| = n$. For any support $S \subseteq \{1, \ldots, n\}$, let $\mathbf{1}_S \in \{0, 1\}^n$ denote its indicator vector. That is, $(\mathbf{1}_S)_i = 1$ if $i \in S$ and $0$ otherwise. Under DR with fixed global thresholds $\tau^+ > \tau^-$, the model must output the exact score vectors

$$\mathbf{z}_S = (\tau^+ - \tau^-)\mathbf{1}_S + \tau^-\mathbf{1}$$

for all supports $S$. This yields the inclusion $\mathrm{span}\{\mathbf{z}_S : S \subseteq [n]\} \subseteq \mathrm{span}\{\mathbf{1}_S : S \subseteq [n]\} + \mathrm{span}\{\mathbf{1}\}$, where $[n]$ denotes the set $\{1, \ldots, n\}$. Since $\mathbf{1} = \mathbf{1}_{[n]}$, we have $\mathrm{span}\{\mathbf{1}_S : S \subseteq [n]\} \supseteq \mathrm{span}\{\mathbf{1}\}$. Therefore, we have simply $\mathrm{span}\{\mathbf{z}_S : S \subseteq [n]\} \subseteq \mathrm{span}\{\mathbf{1}_S : S \subseteq [n]\}$.

To obtain the reverse inclusion, we write the equality

$$\mathbf{1}_S = \frac{1}{\tau^+ - \tau^-}(\mathbf{z}_S - \tau^-\mathbf{1})$$

which gives $\mathrm{span}\{\mathbf{1}_S : S \subseteq [n]\} \subseteq \mathrm{span}\{\mathbf{z}_S : S \subseteq [n]\} + \mathrm{span}\{\mathbf{1}\}$. Notice that $\mathbf{1} \propto \mathbf{z}_{[n]} = \tau^+\mathbf{1}$ and therefore $\mathrm{span}\{\mathbf{z}_S : S \subseteq [n]\} \supseteq \mathrm{span}\{\mathbf{1}\}$. This leads to the reverse inclusion $\mathrm{span}\{\mathbf{1}_S : S \subseteq [n]\} \subseteq \mathrm{span}\{\mathbf{z}_S : S \subseteq [n]\}$.

Putting the two inclusions together, we get $\mathrm{span}\{\mathbf{z}_S : S \subseteq [n]\} = \mathrm{span}\{\mathbf{1}_S : S \subseteq [n]\}$. And since the singleton indicators $\mathbf{1}_{\{i\}}$ are the standard basis vectors, $\mathrm{span}(\{\mathbf{1}_S : S \subseteq [n]\}) = \mathbb{R}^n$. So, to span the space of all $\mathbf{z}_S$, the bottleneck dimension must satisfy $d \geq n$.

**Support-specific thresholds.** In this setting, the thresholds $\tau_S^+, \tau_S^-$ may vary per support. For any support $S$ and its corresponding indicator vector $\mathbf{1}_S$, the target score vector is valid if it matches $\mathbf{1}_S$ up to an arbitrary affine transformation (scaling and shifting)

$$\mathbf{z}_S' = \alpha_S \mathbf{1}_S + \beta_S \mathbf{1}$$

where $\alpha_S \neq 0$ is the support-specific margin and $\beta_S$ is the offset. This means that the model is only required to produce $\mathbf{z}_S$ (from the fixed thresholds case) modulo adding a scalar multiple of the all-ones vector $\mathbf{1}$. As such, we work in the quotient space $\mathbb{R}^n / \mathrm{span}(\mathbf{1})$ which has dimension $n - 1$ (same as for softmax). Therefore it suffices that $d \geq n - 1$. $\qquad\square$

**Theorem 3.6** (DR realisability). *Realising the exact uniform distributions (with fixed or support-specific thresholds) for a graph $\mathbf{Y}$ requires a model rank $d \geq \mathrm{rank}(\mathbf{Y}) - 1$.*

*Proof.* **Fixed thresholds.** Consider a perfect solution $\mathbf{Z}^\star$ which assigns a uniform high score $\tau_+$ to true triples and a uniform low score $\tau_-$ to false ones. The solution should satisfy $\mathbf{Y} = \frac{1}{\tau_+ - \tau_-}(\mathbf{Z}^\star - \tau_-\mathbf{1})$, where $\mathbf{1}$ is a matrix of ones with the same size as $\mathbf{Y}$. Since $\mathbf{Z}^\star$ has rank at most $d$ and $\tau_-\mathbf{1}$ is a matrix of rank at most 1, the equality is only possible if $\mathrm{rank}(\mathbf{Y}) \leq d + 1$.

**Support-specific thresholds.** Each score vector $\mathbf{z}_{s,r,:}^\star$ must be constant on the true object set $S$ and constant on its complement, but the two constants may depend on $(s, r)$. That is, $z_{i,o}^\star = \tau_i^+$ if $y_{i,o} = 1$ and $z_{i,o}^\star = \tau_i^-$ if $y_{i,o} = 0$ where $\tau_i^+ > \tau_i^-$ are row-dependent scalars. For each row $i$, subtracting the (row-dependent) offset $\tau_i^-$ gives the relative form

$$z_{i,o}^\star - \tau_i^- = (\tau_i^+ - \tau_i^-)y_{i,o}.$$

Let $\mathbf{t}^-$ denote the vector of $\tau_i^-$ values and $\mathbf{t}^-\mathbf{1}^\top$ the matrix whose i-th row equals $\tau_i^-\mathbf{1}^\top$. The above equality in matrix form is

$$\mathbf{Z}^\star - \mathbf{t}^-\mathbf{1}^\top = \mathrm{diag}(\tau_i^+ - \tau_i^-)\,\mathbf{Y}.$$

The matrix on the left has rank at most $d+1$, because $\mathbf{Z}^\star$ has rank at most $d$ and $\mathbf{t}^-\mathbf{1}^\top$ has rank 1. The matrix on the right is obtained from $\mathbf{Y}$ by left-multiplying with a diagonal matrix, which does not change its rank. Therefore, $\mathrm{rank}(\mathbf{Y}) \leq d+1$. $\qquad\square$

**Theorem 4.1.** *Let* $\mathbf{Y}^\pm \in \{-1,+1\}^{|\mathcal{E}||\mathcal{R}|\times|\mathcal{E}|}$ *be the sign matrix representing a KG with subject-relation pairs on the first axis and objects on the second.* $\mathbf{Y}^\pm$ *can be exactly decomposed as* $\mathbf{Y}^\pm = \mathrm{sign}(\mathbf{H}\mathbf{E}^\top)$ *where* $\mathrm{sign}$ *is applied element-wise,* $\mathbf{H} \in \mathbb{R}^{|\mathcal{E}||\mathcal{R}|\times(2c+1)}$, $\mathbf{E} \in \mathbb{R}^{|\mathcal{E}|\times(2c+1)}$, *and* $c$ *is the maximum out-degree across all subject-relation pairs in the KG.*

*Proof.* We extend the approach of Chanpuriya et al. (2020); Alon et al. (1985) to the case of rectangular bipartite adjacency matrices $\mathbf{A}^\pm \in \{-1,+1\}^{N\times M}$, where $N$ represents the number of source nodes and $M$ the number of destination nodes. Unlike the usual boolean adjacency matrices, we use $+1$ (or simply $+$) to denote the existence of an edge and $-1$ (or simply $-$) to denote the absence of an edge. $c$ denotes the maximum out-degree of any source node. For the setting of Theorem 4.1, we take $N = |\mathcal{E}||\mathcal{R}|$ and $M = |\mathcal{E}|$.

Our goal is to decompose $\mathbf{A}^\pm$ by expressing each row $\mathbf{a}_i \in \{-1,+1\}^M$ through the equation

$$\mathrm{sign}(\mathbf{V}\mathbf{x}_i) = \mathbf{a}_i,$$

where

- sign is the sign function applied element-wise,

- $\mathbf{V} \in \mathbb{R}^{M\times(2c+1)}$ is a Vandermonde matrix – that is, a matrix with $2c+1$ geometric progressions for $M$ variables – with $M$ rows associated to the destination nodes, and

- $\mathbf{x}_i \in \mathbb{R}^{2c+1}$ is a vector of polynomial coefficients associated with source node $i$.

That is, $\mathbf{x}_i$ define the coefficients of a polynomial of degree $2c$ which, evaluated at each destination node $t = 1, \ldots, M$, yields a score $p_i(t)$ that determines the sign of the edge between node $i$ and node $t$.

If we successfully find such coefficients $\mathbf{x}_i$ for each source node $i = 1, \ldots, N$ – which will be guaranteed by the degree $2c$ of the polynomial –, then we can reconstruct the entire adjacency matrix $\mathbf{A}^\pm$ as

$$\mathbf{A}^\pm = \mathrm{sign}(\mathbf{V}\mathbf{X}^\top),$$

where $\mathbf{X} \in \mathbb{R}^{N\times(2c+1)}$ is the matrix of coefficients $\mathbf{x}_i$ for all source nodes stacked as rows.

**Constructing the Vandermonde Matrix** We define the Vandermonde matrix $\mathbf{V}$ such that

$$v_{t,j} = t^{j-1}, \quad t = 1, \ldots, M, \; j = 1, \ldots, 2c+1.$$

That is, evaluating $\mathbf{V}\mathbf{x}_i \in \mathbb{R}^M$ means evaluating the $2c$-degree polynomial $p_i(t)$ at the integers $t = 1, \ldots, M$, each integer $t$ representing a destination node.

**Fitting the Polynomials** Next, we wish to construct the coefficients $\mathbf{x}_i$ such that the polynomial $p_i(t)$ matches the adjacency pattern $\mathbf{a}_i$:

- $p_i(t) > 0$ for indices where $a_{i,t} = +1$ and

- $p_i(t) < 0$ for indices where $a_{i,t} = -1$.

To achieve this, we choose $p_i(t)$ to have roots at appropriate locations in the intervals between the integer points.

Consider for example the pattern $\mathbf{a}_i = [-\quad-\quad+\quad-\quad+\quad-]$. We need $p_i(t)$ to be positive for $t = 3$ and $t = 5$ and negative for all other $t$. We can choose $p_i(t)$ to have two roots at $t = 3 \pm \epsilon$ and

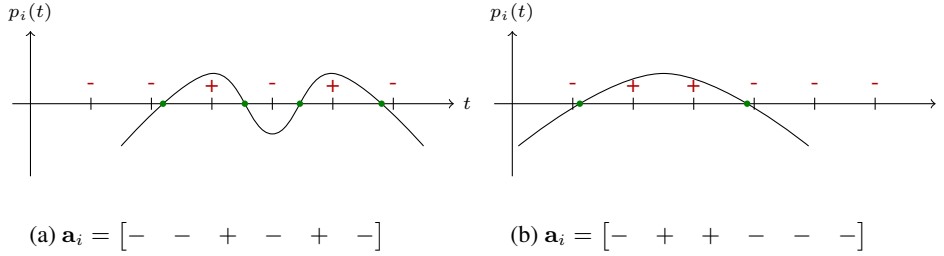

(a) $\mathbf{a}_i = \begin{bmatrix} - & - & + & - & + & - \end{bmatrix}$      (b) $\mathbf{a}_i = \begin{bmatrix} - & + & + & - & - & - \end{bmatrix}$

Figure 4: Fitting a polynomial $p_i(t)$ to the adjacency sign pattern $\mathbf{a}_i$. The sign of $p_i(t)$ at each integer $t = 1, \ldots, M$ indicates the existence of an edge from $i$ to $t$. **(a) For any adjacency pattern with $c$ connections, there exists a polynomial of degree $2c$ that represents the pattern.** This polynomial has $2c$ roots, shown as green dots, which isolate the $c$ points where $p_i(t)$ is positive. **(b) With a better ordering of the ones in $\mathbf{a}_i$, the polynomial may require a lower degree.** Fewer sign changes are needed if the ones in $\mathbf{a}_i$ form blocks.

two roots at $t = 5 \pm \epsilon$, where $0 < \epsilon < 1$, and ensuring that the polynomial has positive signs where needed. This is illustrated in Figure 4a.

In general, if the polynomial has degree $2c$, we can always find $2c$ roots that isolate the (at most $c$) points where $p_i(t)$ is positive. As a side note, notice that this degree requirement is an upper bound. *In practice, the bound can be made tighter depending on the ordering of the entities in the adjacency matrix.* In some cases, the ones in $\mathbf{a}_i$ may appear consecutively (forming *blocks*) so that fewer sign changes are needed. This is particularly likely for graphs with high clustering coefficients, in which case each cluster can be grouped in the ordering (Figure 4b). Finding the optimal ordering is a hard problem and is dataset-specific. Intuitively, datasets that follow power-law degree distributions should be easier to fit, because the ordering needs to be optimised only for the few entities that have many connections. *This is why the average out-degree also gives an important hint on the difficulty of the dataset.*

**Reconstructing the Adjacency Matrix**    Once we have determined a coefficient vector $\mathbf{x}_i$ for each source node $i$, we construct the matrix $\mathbf{X} \in \mathbb{R}^{N \times (2c+1)}$ by stacking them as rows. Then, the entire adjacency matrix can be written as $\mathbf{A}^{\pm} = \text{sign}(\mathbf{V}\mathbf{X}^{\top})$, yielding the desired result.

**Interpreting the Embeddings**    This decomposition suggests a natural interpretation in terms of node embeddings. Each destination node $t$ corresponds to an object $o$ and is embedded as the vector

$$\mathbf{v}_t = \begin{bmatrix} 1 & t & t^2 & \cdots & t^{2c} \end{bmatrix},$$

corresponding to evaluating the polynomial basis at $t$. Each source node $i$ corresponds to a subject-relation pair $(s, r)$ and is embedded as the coefficient vector $\mathbf{x}_i$ defining the polynomial $p_i(t)$. The dot product $\mathbf{v}_t^{\top} \mathbf{x}_i = p_i(t)$ determines the sign at $(i, t)$, i.e., the existence of the triple $(s, r, o)$.

$\square$

## C   SUBJECT PREDICTION

Subject prediction is the task of predicting suitable subject entities for $(?, r, o)$ queries. There are conventionally two main approaches to subject prediction with KGE models.

**Vector-matrix multiplication**    Bilinear models (see Section 2.2.1) can be used to score all subject entities against a relation-object pair as a vector-matrix multiplication:

$$\mathbf{z}_{:,r,o} = \mathbf{E}\mathbf{h}_{r,o} \tag{4}$$

where $\mathbf{h}_{r,o} \in \mathbb{R}^d$ is an embedded representation of the relation-object pair $(r, o)$ and $\mathbf{E} \in \mathbb{R}^{|\mathcal{E}| \times d}$ is the embedding matrix of all subject entity candidates. For example, in RESCAL's score function $\mathbf{e}_s^{\top} \mathbf{W}_r \mathbf{e}_o$ Nickel et al. (2011), we have $\mathbf{h}_{r,o} = \mathbf{W}_r \mathbf{e}_o$, where $\mathbf{W}_r \in \mathbb{R}^{d \times d}$ is a matrix of shared parameters for relation $r$. In the score function $(\mathbf{e}_s \odot \mathbf{w}_r)^{\top} \mathbf{e}_o$ of DISTMULT (Yang et al., 2014), we

have $\mathbf{h}_{r,o} = \mathbf{w}_r \odot \mathbf{e}_o$, where $\odot$ is the element-wise (Hadamard) product, and $\mathbf{e}_s, \mathbf{w}_r, \mathbf{e}_o \in \mathbb{R}^d$. Under this perspective, our theoretical results on rank bottlenecks in KGEs for solving object prediction (Section 3) can be extended to subject prediction by replacing the hidden state $\mathbf{h}_{s,r}$ with $\mathbf{h}_{r,o}$. But, our KGE-MoS solution is not applicable simultaneously for both object and subject prediction as the mixture of softmaxes breaks the bilinearity of the score function.

**Inverse relations** The most general approach which applies to all KGE methods – including neural network methods – is to introduce inverse relations $r^{-1}$ and to convert the subject prediction query $(?, r, o)$ into an object prediction query $(o, r^{-1}, ?)$ (Dettmers et al., 2017; Lacroix et al., 2018). This introduces $|\mathcal{R}|$ new relations and embeddings, but this parameter count is often negligible compared to the number of parameters of the entities.

$$\mathbf{z}_{:,r,o} := \mathbf{z}_{o,r^{-1},:} = \mathbf{h}_{o,r^{-1}} \mathbf{E}^\top \tag{5}$$

In this case, both the theoretical results and the KGE-MoS solution are immediately applicable. We can use the same KGE-MoS layer as follows:

$$P(S|o, r^{-1}) = \sum_{k=1}^{K} \pi_{o,r^{-1},k} \,\mathrm{softmax}(f_k(\mathbf{h}_{o,r^{-1}})\mathbf{E}^\top), \quad \sum_{k=1}^{K} \pi_{o,r^{-1},k} = 1 \tag{6}$$

where $S$ is a random variable that ranges over the same entities as $O$.

# D    BACKGROUND (CONT.)

## D.1    OBJECT PREDICTION AS A VECTOR-MATRIX MULTIPLICATION

In this section, we elaborate on how to rewrite the different KGE models as a function linear in the object embedding $\phi(s, r, o) = \mathbf{h}_{s,r}^\top \mathbf{e}_o$, with $\mathbf{h}_{s,r}, \mathbf{e}_o \in \mathbb{R}^d$.

**Bilinear models** In RESCAL's score function $\mathbf{e}_s^\top \mathbf{W}_r \mathbf{e}_o$ Nickel et al. (2011), we have $\mathbf{h}_{s,r}^\top = \mathbf{e}_s^\top \mathbf{W}_r$, where $\mathbf{W}_r \in \mathbb{R}^{d \times d}$ is a matrix of shared parameters for relation $r$. In the score function $(\mathbf{e}_s \odot \mathbf{w}_r)^\top \mathbf{e}_o$ of DISTMULT (Yang et al., 2014), we have $\mathbf{h}_{s,r}^\top = (\mathbf{e}_s \odot \mathbf{w}_r)^\top$, where $\odot$ is the element-wise (Hadamard) product, and $\mathbf{e}_s, \mathbf{w}_r, \mathbf{e}_o \in \mathbb{R}^d$. The same can be derived for COMPLEX (Trouillon et al., 2017), an extension of DISTMULT to complex-valued embeddings to handle asymmetry in the score function, or for CP (Lacroix et al., 2018) and SIMPLE (Kazemi & Poole, 2018), which use different embedding spaces for subjects and objects – though COMPLEX and SIMPLE with embedding sizes $d$ give forms $\mathbf{h}_{s,r}^\top \mathbf{e}_o$ with bottleneck dimension $2d$ as we detail in the next paragraph. Finally, TUCKER (Balažević et al., 2019) decomposes the score tensor $\boldsymbol{\mathcal{Z}} \in \mathbb{R}^{|\mathcal{E}| \times |\mathcal{R}| \times |\mathcal{E}|}$ as $\boldsymbol{\mathcal{Z}} = \boldsymbol{\mathcal{W}} \times_1 \mathbf{E} \times_2 \mathbf{R} \times_3 \mathbf{E}$ where $\boldsymbol{\mathcal{W}} \in \mathbb{R}^{d \times d_r \times d}$ is a core tensor of shared parameters across all entities and relations and $\times_n$ are mode-$n$ products.[2] $\mathbf{E} \in \mathbb{R}^{|\mathcal{E}| \times d}$, $\mathbf{R} \in \mathbb{R}^{|\mathcal{R}| \times d_r}$ are entity and relation embeddings, respectively. The score function is then $\phi_{\text{TUCKER}}(s, r, o) = \boldsymbol{\mathcal{W}} \times_1 \mathbf{e}_s \times_2 \mathbf{w}_r \times_3 \mathbf{e}_o$. Let $\mathbf{h}_{s,r} = \boldsymbol{\mathcal{W}} \times_1 \mathbf{e}_s \times_2 \mathbf{w}_r \in \mathbb{R}^d$. Then, $\phi_{\text{TUCKER}}(s, r, o) = (\mathbf{h}_{s,r}^{\text{TUCKER}})^\top \mathbf{e}_o$. Specific implementations of the core tensor $\boldsymbol{\mathcal{W}}$ recovers the other bilinear models as special cases.

**COMPLEX, SIMPLE** As explained by Kazemi & Poole (2018); Balažević et al. (2019), COMPLEX (Trouillon et al., 2017) can be seen as a special case of TUCKER and bilinear models by considering the real and imaginary part of the embedding concatenated in a single vector, e.g., $[\mathrm{Re}(\mathbf{e}_o); \mathrm{Im}(\mathbf{e}_o)] \in \mathbb{R}^{2d}$ for the object. We detail this to highlight that the rank bottleneck for these models is $2d$ rather than $d$. The score function of COMPLEX is

$$\phi_{\text{COMPLEX}}(s, r, o) = (\mathrm{Re}(\mathbf{e}_s) \odot \mathrm{Re}(\mathbf{w}_r))^\top \mathrm{Re}(\mathbf{e}_o) + (\mathrm{Im}(\mathbf{e}_s) \odot \mathrm{Re}(\mathbf{w}_r))^\top \mathrm{Im}(\mathbf{e}_o)$$
$$(\mathrm{Re}(\mathbf{e}_s) \odot \mathrm{Im}(\mathbf{w}_r))^\top \mathrm{Im}(\mathbf{e}_o) - (\mathrm{Im}(\mathbf{e}_s) \odot \mathrm{Im}(\mathbf{w}_r))^\top \mathrm{Re}(\mathbf{e}_o)$$

Let us define $\mathbf{h}_{sr}^1 = \mathrm{Re}(\mathbf{e}_s) \odot \mathrm{Re}(\mathbf{w}_r) - \mathrm{Im}(\mathbf{e}_s) \odot \mathrm{Im}(\mathbf{w}_r)$ and $\mathbf{h}_{sr}^2 = \mathrm{Im}(\mathbf{e}_s) \odot \mathrm{Re}(\mathbf{w}_r) + \mathrm{Re}(\mathbf{e}_s) \odot \mathrm{Im}(\mathbf{w}_r)$, with $\mathbf{h}_{sr}^1, \mathbf{h}_{sr}^2 \in \mathbb{R}^d$. We have

$$\phi_{\text{COMPLEX}}(s, r, o) = (\mathbf{h}_{sr}^1)^\top \mathrm{Re}(\mathbf{e}_o) + (\mathbf{h}_{sr}^2)^\top \mathrm{Im}(\mathbf{e}_o).$$

---

[2] The mode-$n$ product $\boldsymbol{\mathcal{X}} \times_n \mathbf{A}$ is the tensor obtained by multiplying each slice of $\boldsymbol{\mathcal{X}}$ along the $n$-th mode by the corresponding column of $\mathbf{A}$. That is, $(\boldsymbol{\mathcal{X}} \times_n \mathbf{A})_{i_1 \dots i_{n-1} j \, i_{n+1} \dots i_N} = \mathbf{a}_{j,:}^\top \mathbf{x}_{i_1, \dots, i_{n-1}, :, i_{n+1}, \dots, i_N}$.

Concatenating the real and imaginary parts of $\mathbf{e}_o$ into a single vector $\mathbf{e}_o^{\text{COMPLEX}} = [\text{Re}(\mathbf{e}_o); \text{Im}(\mathbf{e}_o)] \in \mathbb{R}^{2d}$, and concatenating the hidden vectors $\mathbf{h}_{sr}^1$ and $\mathbf{h}_{sr}^2$ into a single vector $\mathbf{h}_{sr}^{\text{COMPLEX}} = [\mathbf{h}_{sr}^1; \mathbf{h}_{sr}^2] \in \mathbb{R}^{2d}$, we have

$$\phi_{\text{COMPLEX}}(s, r, o) = (\mathbf{h}_{sr}^{\text{COMPLEX}})^\top \mathbf{e}_o^{\text{COMPLEX}}$$

which is linear in the object embedding $\mathbf{e}_o^{\text{COMPLEX}}$, this time with bottleneck dimension $2d$. A similar trivial result can be done for SIMPLE Kazemi & Poole (2018), again considering the concatenation of two embeddings in a single vector in $\mathbb{R}^{2d}$.

**Neural KGEs** CONVE (Dettmers et al., 2017) scores triples as $f(\text{vec}(f([\mathbf{e}_s; \mathbf{w}_r] * \mathbf{w}))\, \mathbf{W})^\top \mathbf{e}_o$, with $f$ a non-linearity, $*$ a convolution operator, and vec a vectorization operator. The score function is linear in the object embedding $\mathbf{e}_o$. Similarly, STARE (Galkin et al., 2020) uses a score function $\text{Linear}(\text{SumPooling}(\text{Transformer}([\mathbf{e}_s + \mathbf{pe}[0]; \mathbf{w}_r + \mathbf{pe}[1]])))^\top \mathbf{e}_o$, where $\mathbf{pe}$ are positional encodings, that has non-linearities in the transformer and pooling constructing $\mathbf{h}_{s,r}$ but that is still linear in the object embedding $\mathbf{e}_o$. R-GCNs (Schlichtkrull et al., 2017) and COMPGCN (Vashishth et al., 2019) use powerful message passing schemes to build node representations for the entities $\mathbf{e}_s$ and $\mathbf{e}_o$, but still use a simple bilinear scoring function to score triples given the resulting representations – DISTMULT's score function giving the best performance. Language-based models like SIMKGC (Wang et al., 2022a) or MEMKGC (Choi et al., 2021) use BERT or other masked language models to encode powerful entity and relation embeddings $\mathbf{e}_s$, $\mathbf{e}_o$ and $\mathbf{w}_r$, but still use a linear projection to score objects in their final layer. COLE (Liu et al., 2022) uses two heads, one transformer and one BERT, and both use a linear output layer to score objects. LMKE (Wang et al., 2022b) is designed for triple classification but has a variant C-LMKE designed for link prediction, which uses a linear output layer where rows correspond to entities embeddings $\mathbf{e}_o$.

### D.2 OUTPUT FUNCTIONS IN KGES

In this section, we further detail the different output functions that KGEs parameterised over the years and the loss functions that were used to train them.

**Raw scores with margin-based loss (RR)** Early KGEs (Bordes et al., 2013; Yang et al., 2014) used margin-based loss functions for ranking reconstruction (RR). The loss ensures that the scores of true triples are higher than the scores of false triples by at least a margin $\gamma > 0$, but does not ensure sign or calibrated reconstruction.

**Sigmoid layers with BCE loss (RR, SR, possibly DR)** Trouillon et al. (2017), followed by Dettmers et al. (2017), proposed to use binary cross-entropy (BCE) loss between the scores and the binary representation of the graph. They define a sigmoid layer $P(y_o = 1 | \mathbf{h}_{s,r}) = \sigma(\mathbf{h}_{s,r}^\top \mathbf{e}_o)$, where $y_o = 1$ if $(s, r, o) \in \mathcal{G}$ and $0$ otherwise. This defines a multi-label classifier with predictions $\hat{\mathbf{y}} = \mathbb{1}(\mathbf{E}\mathbf{h}_{s,r} > 0) \in \{0, 1\}^{|\mathcal{E}|}$ where $\mathbb{1}$ is the indicator function applied element-wise and the $i$-th entry is the binary prediction for object $i$. This classifier naturally aligns with sign-label reconstruction (SR). Calibrated scores (DR) were not a concern in this setting until Arakelyan et al. (2021; 2023) started to combine the binary prediction scores of several simple queries to answer complex queries, which requires the scores to be calibrated.

**Softmax layers with CE loss (RR, possibly DR)** Kadlec et al. (2017) proposed to use a cross-entropy (CE) loss. For a query $(s, r, ?)$, they define a vector of probabilities $\mathbf{p}_{s,r} = \text{softmax}(\mathbf{E}\mathbf{h}_{s,r}) \in \Delta^{|\mathcal{E}|}$, with $\Delta^{|\mathcal{E}|}$ the probability simplex over the objects. Here, softmax is applied element-wise to get the components $p_{s,r,o} = \exp(\mathbf{h}_{s,r}^\top \mathbf{e}_o) / \sum_{o' \in \mathcal{E}} \exp(\mathbf{h}_{s,r}^\top \mathbf{e}_{o'})$. This vector should align with the data distribution (labels of the corresponding triples, normalised to sum to 1). Although softmax is typically used for predicting a single correct class, here it is repurposed for multi-label scenarios by retrieving top-$k$ entries and ranking likely completions (RR). This modeling also opens the door to sampling (Loconte et al., 2023) which requires calibrated scores (DR).

In comprehensive benchmarking efforts, Ruffinelli et al. (2020) and Ali et al. (2022) re-evaluated all score functions with all possible output functions and loss functions for ranking reconstruction (RR). They found that softmax-based modeling generally outperform other approaches, closely followed

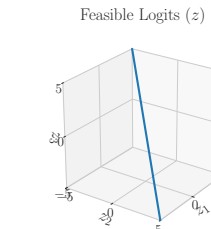 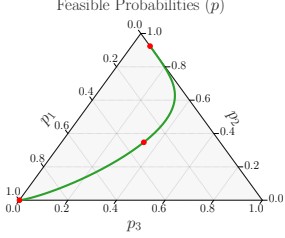 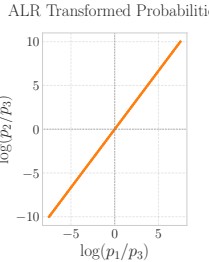

Figure 5: Visualisation of the softmax bottleneck. The blue line is the linear subspace of feasible scores for a fixed matrix $\mathbf{E}$, where each point is calculated for a different hidden state $\mathbf{h}_{s,r}$. The green line is the set of feasible probability distributions after the softmax function. The orange line is the additive-log ratio of the probabilities. **The feasible distributions are constrained to a manifold that is linear in the ALR space.** Any groundtruth that does not respect this constraint cannot be represented by the model.

by modeling with BCE loss. Models trained with margin-based loss functions were consistently the worst performing. Note that the authors did not evaluate the performance of models for sign-label reconstruction (SR) or calibrated score reconstruction (DR).

# E   RANK BOTTLENECKS IN KGEs (CONT.)

## E.1   LOG-LINEARITY OF THE SOFTMAX BOTTLENECK

As we mention in 3.3, the softmax function maps the $d$-dimensional linear subspace of scores to a smooth $d$-dimensional manifold within the $(|\mathcal{E}|-1)$-dimensional probability simplex $\Delta^{|\mathcal{E}|}$. While this manifold is non-linear, its shape is relatively simple and can be described by the following constraint: its composition with the log function is a linear function of the scores. An algebraic perspective on this bottleneck is provided by Yang et al. (2017). Let $\mathbf{A} \in \mathbb{R}^{|\mathcal{E}||\mathcal{R}| \times |\mathcal{E}|}$ be the log-probability matrix where $a_{i,j} = \log P(O = j | \mathbf{h}_i)$. We have

$$\mathbf{A} = \begin{bmatrix} \log \text{softmax}(\mathbf{Eh}_{1,1}) \\ \vdots \\ \log \text{softmax}(\mathbf{Eh}_{|\mathcal{E}|,|\mathcal{R}|}) \end{bmatrix} = \mathbf{HE}^\top - \begin{bmatrix} c_{1,1} \\ \vdots \\ c_{|\mathcal{E}|,|\mathcal{R}|} \end{bmatrix} \begin{bmatrix} 1 & \cdots & 1 \end{bmatrix}$$

where $c_i = \log \sum_{j=1}^{|\mathcal{E}|} \exp(\mathbf{Eh}_i)_j \in \mathbb{R}$ is the log-partition function. Since $\mathbf{HE}^\top$ has a rank at most $d$, and the term involving log-partition functions has a rank at most 1, the resulting log-probability matrix $\mathbf{A}$ has a rank of at most $d + 1$.

We can visualise this log-linearity using log-ratio transformations. For example, for a vector of probabilities $\mathbf{p} \in \Delta^{|\mathcal{E}|}$, we use an Additive Log-Ratio (ALR) transformation with $p_{|\mathcal{E}|}$ as the reference:

$$\text{ALR}(\mathbf{p}) = \begin{bmatrix} \log \frac{p_1}{p_{|\mathcal{E}|}} \\ \vdots \\ \log \frac{p_{|\mathcal{E}|-1}}{p_{|\mathcal{E}|}} \end{bmatrix} \tag{7}$$

This transformation maps the simplex to a $d - 1$ dimensional space where the log-partitions are eliminated in the ratios. As we visualise in Figure 5, the log-ratio of the probabilities span a linear subspace of the possible probability ratios. Any groundtruth that does not respect this log-linear constraint cannot be represented by the model.

## E.2   RANKING REALISABILITY BOUNDS FOR BILINEAR KGEs FROM WANG ET AL. (2017)

As explained in Section 3.1, Theorem 3.2 gives a condition

$$d_{\text{RR}} \geq \text{srank}(\mathbf{Y}^\pm) - 1$$

for realising the exact rankings of a graph $\mathbf{Y}$ that lower bounds any KGE-specific condition. Wang et al. (2017) provide sufficient bounds for some specific bilinear KGEs. The tightest are for RESCAL and COMPLEX, which have the sufficient bound

$$d^+_{\text{RESCAL}} = d^+_{\text{COMPLEX}} = 2 \sum_{r \in \mathcal{R}} \text{rrank}(\mathbf{Y}_r)$$

where

- $\mathbf{Y}_r \in \{0, 1\}^{|\mathcal{E}| \times |\mathcal{E}|}$ is the adjacency matrix for relation $r$.
- $\text{rrank}(\mathbf{Y}_r)$ is the rounding rank of $\mathbf{Y}_r$ with threshold 0.5, which is the minimum rank of any real matrix $\mathbf{M}$ with $\text{round}(\mathbf{M}) = \mathbf{Y}_r$, where $\text{round}(x) = \mathbb{1}[x > 0.5]$ is applied element-wise and $\mathbb{1}[\cdot]$ is the indicator function.

Notice that the bound is a sufficiency bound, not a necessary one. The necessary and sufficient bound for RESCAL and COMPLEX are somewhere in between $d_{\text{RR}}$ and $d^+_{\text{RESCAL}}$ and are still unknown. Next, we confirm that this bound is (up to $2|\mathcal{R}|$ times) larger than the one in Theorem 3.2.

**Relating sign ranks and rounding ranks**   Neumann et al. (2016) show that the rounding rank of any boolean matrix $\mathbf{A} \in \{0, 1\}^{n \times m}$ and sign rank of the corresponding sign matrix $\mathbf{A}^{\pm} = 2\mathbf{A} - 1 \in \{-1, +1\}^{n \times m}$ are tightly related. In particular, with the threshold 0.5 it can be shown that $\text{rrank}(\mathbf{A})$ and $\text{srank}(\mathbf{A}^{\pm})$ differ by at most 1:

1. Any matrix $\mathbf{B}$ with $\text{round}(\mathbf{B}) = \mathbf{A}$ defines a sign matrix $\mathbf{C} := B - \frac{1}{2}\mathbf{J}$ (where $\mathbf{J}$ is the all-ones matrix) that satisfies $\text{sign}(\mathbf{C}) = \mathbf{A}^{\pm}$. Therefore, if the rank of $\mathbf{B}$ is $r$, the the rank of $\mathbf{C}$ is at most $r + 1$ (sum of $B$ and a rank-1 matrix). So $\text{srank}(\mathbf{A}^{\pm}) \leq \text{rrank}(\mathbf{A}) + 1$.

2. Conversely, any matrix $\mathbf{C}$ with $\text{sign}(\mathbf{C}) = \mathbf{A}^{\pm}$ defines a matrix $\mathbf{B} := \frac{1}{2}\mathbf{J} + \epsilon\mathbf{C}$ with $\epsilon > 0$ that satisfies $\text{round}(\mathbf{B}) = \mathbf{A}$. Therefore, if the rank of $\mathbf{C}$ is $r$, the the rank of $\mathbf{B}$ is at most $r + 1$ (sum of a rank-1 matrix and $\mathbf{C}$). So $\text{rrank}(\mathbf{A}) \leq \text{srank}(\mathbf{A}^{\pm}) + 1$.

Therefore, $|\text{rrank}(\mathbf{A}) - \text{srank}(\mathbf{A}^{\pm})| \leq 1$.

**Expressing $d^+_{\text{RESCAL}}$ and $d^+_{\text{COMPLEX}}$ in terms of the sign rank**   Based on the above relationship between rounding ranks and sign ranks, we can express the bound from Wang et al. (2017) as

$$d^+_{\text{RESCAL}} = d^+_{\text{COMPLEX}} = 2 \sum_{r \in \mathcal{R}} \left( \text{srank}(\mathbf{Y}^{\pm}_r) - 1 \right)$$

where $\mathbf{Y}^{\pm}_r = 2\mathbf{Y}_r - 1 \in \{-1, +1\}^{|\mathcal{E}| \times |\mathcal{E}|}$ is the sign matrix for relation $r$.

**Relationship with $d_{\text{RR}}$**   Next, we relate $\sum_{r \in \mathcal{R}} \text{srank}(\mathbf{Y}^{\pm}_r)$ to $\text{srank}(\mathbf{Y}^{\pm})$. $\mathbf{Y}^{\pm}$ can be written as the block matrix stacking each relation matrix $\mathbf{Y}^{\pm} = \begin{bmatrix} \mathbf{Y}^{\pm}_1 \\ \vdots \\ \mathbf{Y}^{\pm}_{|\mathcal{R}|} \end{bmatrix}$. Then, $\text{srank}(\mathbf{Y}^{\pm}) = \sum_{r \in \mathcal{R}} \text{srank}(\mathbf{Y}^{\pm}_r)$ only when all rows in all relation matrices are linearly independent. However, this might not be the case in practice. In the degenerate case where all the relation matrices are identical, we have $\text{srank}(\mathbf{Y}^{\pm}) = \text{srank}(\mathbf{Y}^{\pm}_1)$, whereas $\sum_{r \in \mathcal{R}} \text{srank}(\mathbf{Y}^{\pm}_r) = |\mathcal{R}| \cdot \text{srank}(\mathbf{Y}^{\pm}_1)$. In that case, the lower bound $d_{\text{RR}} \geq \text{srank}(\mathbf{Y}^{\pm}_1) - 1$ can be roughly $2|\mathcal{R}|$ times smaller than $d^+_{\text{RESCAL}} = d^+_{\text{COMPLEX}} = 2|\mathcal{R}|(\text{srank}(\mathbf{Y}^{\pm}_1) - 1)$.

# F   EXPERIMENTS

## F.1   DATASETS

We use the usual splits for FB15K237 Toutanova & Chen (2015), WN18RR Dettmers et al. (2017) and ogbl-biokg Hu et al. (2020). openbiolink Breit et al. (2020) comes with four available datasets. We use the high-quality, directed set downloadable from the PyKeen library Ali et al. (2020),

which filters out test entities that do not appear in the training set and are not learnable by knowledge graph embedding models. `Hetionet` Himmelstein et al. (2017) does not come with pre-defined splits. We obtain the splits via the PyKeen library using a seed of 42. Table 5 reports statistics for all datasets.

Table 5: Dataset statistics and $(s, r, \cdot)$ out-degrees. The sufficient bound $d_{\mathrm{SR}}^+$ for exact sign reconstruction is calculated based on the maximum out-degree according to Theorem 4.1.

| Dataset | #Entities | #Rels | #Triples | Inverse Relations | Out-Degree | | | $d_{\mathrm{SR}}^+$ |
| | | | | | Mean | Median | Max | |
|---|---|---|---|---|---|---|---|---|
| FB15k-237 | 14,541 | 237 | 310,116 | ✗ | 3.03 | 1 | 954 | 1,909 |
| | | | | ✓ | 3.83 | 1 | 4,364 | 8,729 |
| WN18RR | 40,943 | 11 | 93,003 | ✗ | 1.40 | 1 | 473 | 947 |
| | | | | ✓ | 1.70 | 1 | 510 | 1,021 |
| Hetionet | 45,158 | 24 | 2,250,197 | ✗ | 29.07 | 7 | 15,036 | 30,073 |
| | | | | ✓ | 21.66 | 6 | 15,036 | 30,073 |
| ogbl-biokg | 93,773 | 51 | 5,088,434 | ✗ | 40.54 | 14 | 29,328 | 58,657 |
| | | | | ✓ | 37.48 | 12 | 29,328 | 58,657 |
| openbiolink | 180,992 | 28 | 4,559,267 | ✗ | 14.47 | 2 | 2,251 | 4,503 |
| | | | | ✓ | 18.12 | 2 | 18,420 | 36,841 |

## F.2 METRICS

**Filtered NLL**   The negative log-likelihood (NLL) metric measures how well a model assigns probability to the true triples in the test set. However, in KGC, because the training and test sets are disjoint, a model trained to assign high probability to training triples might leave little probability mass for test triples, even if they follow similar patterns. In this case, the NLL can unfairly penalise models that fit the training set well but are able to generalise out-of-distribution, as generalisation in KGC is always *out-of-distribution* rather than in-distribution.

To address this issue, we use a filtered version of NLL when evaluating on the test set. For a $(s, r, ?)$ query, we zero out the probabilities of ground truth objects for that query seen during training, renormalising the probabilities of the model predictions. Formally, given a model prediction $P(o|s, r)$, we define the filtered test probability as

$$P^{\text{filtered}}(o|s, r) = \frac{\mathbb{1}[(s, r, o) \notin \mathcal{G}_{\text{train}}] P(o|s, r)}{\sum_{o' \in \mathcal{E}} \mathbb{1}[(s, r, o') \notin \mathcal{G}_{\text{train}}] P(o'|s, r)} \tag{8}$$

This filtered approach provides a more meaningful evaluation of the model's ability to generalise to unseen triples, as it focuses on the model's ability to distinguish test triples from truly negative samples rather than from training triples that the model has already learned. It prevents situations where a model that accurately assigns high probability to training instances might be unfairly disadvantaged compared to a less informed model when evaluating predictions on the disjoint test set (Figure 6).

**Ranking metrics**   The quality of a KGC model is commonly assessed by ranking the scores of test triples $(s, r, o) \in \mathcal{G}_{\text{test}}$. For example, in the context of object prediction, the ranks are computed as

$$R_\theta(s, r, o) := 1 + \sum_{e \in \mathcal{E}'} \mathbb{1}[\Phi_\theta(s, r, e) > \Phi_\theta(s, r, o)]. \tag{9}$$

where $\mathbb{1}[\cdot]$ is the indicator function and the set $\mathcal{E}' := \{e \in \mathcal{E} \mid (s, r, e) \notin \mathcal{G}\}$ filters the rank by only including object entities that do not form KG triples. A pessimistic version of the rank is sometimes considered by taking a non-strict inequality The mean reciprocal rank (MRR) is the mean of the reciprocal ranks MRR $= \frac{1}{|\mathcal{G}_{\text{test}}|} \sum_{(s,r,o) \in \mathcal{G}_{\text{test}}} R_\theta(s, r, o)^{-1}$. The mean rank (MR) is the arithmetic mean of the ranks, but is notably sensitive to outliers. The Hits@k metric is the proportion of test triples that are ranked within the top $k$ positions.

Notice that in `ogbl-biokg`, each test object for $(s, r, ?)$ are not ranked against all entities, but only against a pre-defined set of 500 entities. The same holds for subject prediction $(?, r, o)$.

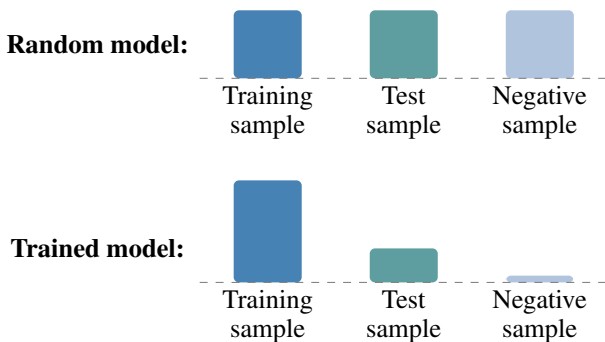

Figure 6: Likelihood assigned by models to training/test/random samples (conceptual graph). Because in KGC the set of training and test samples are disjoint, trained models leave less probability mass for test samples even if the model would assign higher probability to test samples if training samples were not options.

### F.3    EXPERIMENTAL SETTING

**General Hyperparameters**    We guide the choice of hyperparameters by guidelines from the original papers of each model, as well as insights from recent benchmarking papers Ruffinelli et al. (2020); Ali et al. (2022). All baseline models are trained using the Adam optimiser Kingma & Ba (2014) with a learning rate of $10^{-4}$, except for DISTMULT and COMPLEX (without KGE-MOS), which use a higher learning rate of $10^{-3}$ due to their shallow architecture. We use a batch size of 1000 for all models as KGEs are more stable with larger batch sizes Ruffinelli et al. (2020), except for RESCAL, RESCAL-MOS, COMPGCN and COMPGCN-MOS, which use a batch size of 500 on `ogbl-biokg` and `openbiolink` for memory constraints. The models are trained for 30 training set epochs, with early stopping according to validation set metrics at a patience of 8 epochs.

**Regularisation**    Following the findings of Ruffinelli et al. (2020), which showed that dropout Srivastava et al. (2014) is more effective than norm-based regularisation for KGEs, we apply a dropout of 0.1 to the encoded representation $\mathbf{h}_{s,r}$ (recalling that models can be represented as $\phi(s, r, o) = \mathbf{h}_{s,r}^{\top}\mathbf{e}_o$). An exception to this is CONVE, which already incorporates dropout within its neural network layers.

**CONVE specific hyperparameters**    For CONVE, we adopt the hyperparameter settings from its original paper Dettmers et al. (2017). The model reshapes the input subject and relation embeddings into two-dimensional "images" (height 8), followed by a two-layer convolutional network with 32 channels and a kernel size of $(3, 3)$. The output is then passed through a fully connected layer to obtain $\mathbf{h}_{s,r}$. Regularisation for CONVE includes dropout on embeddings (0.2), feature maps after convolution (0.3), and the output of the fully connected layer (0.3). Each layer in CONVE is also followed by a batch normalisation layer Ioffe & Szegedy (2015).

**COMPGCN specific hyperparameters**    For COMPGCN, we use the following hyperparameters: initial embedding dimension of 100 for entities and relations and GCN dimension (which dictates the bottleneck) of $d = 200$, $d = 100$ or $d = 50$ depending on the dataset. We use a single GCN layer without bias terms and use COMPGCN without basis decomposition. Dropout is applied at two stages: 0.1 dropout in the GCN layer and 0.1 hidden dropout after the GCN layer. For the composition operation in COMPGCN, we use DISTMULT as was empirically shown to be the best option in the original paper Vashishth et al. (2019).

**KGE-MOS details and hyperparameters**    For the KGE-MOS layer, defined as:

$$P(O|s,r) = \sum_{k=1}^{K} \pi_{s,r,k} \, \text{softmax}(f_k(\mathbf{h}_{s,r})\mathbf{E}^{\top}), \quad \sum_{k=1}^{K} \pi_{s,r,k} = 1 \qquad (10)$$

we use two-layer projections for $f_k(\mathbf{h}_{s,r})$, maintaining the same intermediate dimension $d$ as the KGE dimension. Each projection layer is followed by batch normalisation Ioffe & Szegedy (2015),

Table 6: Empirical rank of the log-probability matrix for DISTMULT-MOS on the test set of FB15k-237.

| Model | Empirical rank |
|---|---|
| DISTMULT-MOS, $K = 1, d = 200$ | 201 |
| DISTMULT-MOS, $K = 4, d = 200$ | 3,651 |
| DISTMULT-MOS, $K = 1, d = 1000$ | 1,001 |
| DISTMULT-MOS, $K = 4, d = 1000$ | 14,541 |

a non-linear activation, and hidden dropout (applied in that order). To encourage the use of all mixture components, we incorporate the entropy of the distribution defined by the mixture weights $(\pi_{s,r,1}, \ldots, \pi_{s,r,K})$ into the loss function. Our hyperparameter search for KGE-MOS involved a random exploration over learning rate ($10^{-2}$ to $10^{-5}$, log-scale), hidden dropout (0.0 to 0.2), activation type (ReLU, LeakyReLU, GeLU, tanh), and entropy regularisation weight ($10^{-7}$ to 1.0, log-scale). The final reported results for KGE-MOS were obtained with a learning rate of $10^{-4}$, hidden dropout of 0.1, LeakyReLU activation, and an entropy regularisation weight of $10^{-3}$.

**Parameters initialisation** Parameters for the KGEs and all model layers are initialised using Xavier uniform initialisation Glorot & Bengio (2010). Repeated runs are performed with different initialisation seeds for statistical significance.

**Hardware** All experiments are run on NVIDIA A10 GPUs with 24GB of memory. Time of execution for each baseline ranges from a couple of hours for FB15K237, the smallest dataset, to 1-2 days for openbiolink, the largest dataset.

### F.4 ADDITIONAL RESULTS

#### F.4.1 STANDARD DEVIATIONS AND HITS@ METRICS

Tables 9, 10, 11, 12 and 13 report the full results for the FB15k-237, Hetionet, ogbl-biokg, openbiolink, and WN18RR datasets, respectively.

Note that results for COMPLEX use halved embedding sizes $d = 100$ and $d = 500$ because it has real and imaginary parts leading to a rank bottleneck of $2d$ (see Appendix D.1).

On the other hand, while results for COMPGCN use $d = 200$ on FB15k-237 and Hetionet and are comparable to the other models, they are conducted with $d = 100$ on ogbl-biokg, and $d = 50$ on openbiolink due to computational constraints.

#### F.4.2 EMPIRICAL RANK

Borrowing the idea of Yang et al. (2017), we confirm empirically that KGE-MOS breaks the rank bottleneck by measuring the rank of the log-probability matrix measured on the test data for $K = 1$ softmax and $K > 1$ softmaxes. That is, we compute the rank of

$$\begin{bmatrix} \log P(O|sr_1) \\ \vdots \\ \log P(O|sr_{N_{\text{test}}}) \end{bmatrix}$$

where $sr_i$ ranges over subject relation pairs in the test set.

We find the empirical ranks in Table 6. When $K = 1$, the rank is as expected $d + 1$ (where $d$ is the rank of the logits and 1 is the rank of the log partition matrix). **Increasing $K$ quickly increases the empirical rank.** In fact, as in FB15k-237, the number of entities $|\mathcal{E}| = 14,951$, the matrix is even **full rank** when $d = 1000$ and $K > 1$. In other words, the bottleneck is effectively broken.

#### F.4.3 INFERENCE TIME

In Table 7 we report inference time results for all methods.

Table 7: Inference and backpropagation time for a single minibatch on `openbiolink` at $d = 1000$. RESCAL and RESCAL-MoS are compared at batch size 500 to fit on device memory. Other models are compared at batch size 1000.

| Model | Time per batch (ms) |
|---|---|
| DISTMULT | 1.34 |
| DISTMULT-MoS ($K = 1$) | 3.30 |
| DISTMULT-MoS ($K = 4$) | 3.69 |
| RESCAL | 1.51 |
| RESCAL-MoS ($K = 1$) | 3.49 |
| RESCAL-MoS ($K = 4$) | 3.42 |
| CONVE | 2.72 |
| CONVE-MoS ($K = 1$) | 4.74 |
| CONVE-MoS ($K = 4$) | 4.60 |

Table 8: Results on `ogbl-biokg` at $d = 5000$. Breaking rank bottlenecks does not improve performance when the dimension is sufficiently large. However, increasing $d$ comes at the cost of sacrificing critical hyperparameters such as batch size due to GPU memory constraints which deteriorates performance.

| Model ($d = 5000$) | NLL | MRR | Param |
|---|---|---|---|
| DISTMULT-MoS, $K = 1$ | 4.54 | 0.79 | 519.4M |
| DISTMULT-MoS, $K = 4$ | 4.54 | 0.79 | 669.5M |

#### F.4.4 BOTTLENECKS IN HIGHER-DIMENSIONAL SETTINGS ON `OGBL-BIOKG`

Next, we target a negative result: we evaluate whether KGE-MoS still shows a significant performance gain when the embedding dimension $d$ is sufficiently large. Indeed, for a sufficiently large dimension $d$ (dataset dependent), the bottleneck impact is not as great. Our results on `FB15K-237` and `WN18RR` (small datasets) aim to demonstrate this. We try to replicate this insight on one of the larger datasets (`ogbl-biokg`, the most densely connected one) with $d = 5000$.

We had to reduce our batch size from 1000 to 200 samples to accommodate for the larger dimension in the GPU memory (24GB). We measure two baselines, (i) DISTMULT-MoS with $K = 1$, (ii) DISTMULT-MoS with $K = 4$. DISTMULT-MoS with $K = 1$ does not break the rank bottleneck, but is included to isolate the effect of adding an asymmetric layer to DISTMULT vs breaking the rank bottleneck (see our paragraph "Mixture Ablation" in Section 6.1).

The results gives us two findings. (i) Breaking the rank bottleneck does not notably improve the performance on `ogbl-biokg` when $d = 5000$(DISTMULT-MoS, $K = 4 \approx$ DISTMULT-MoS, $K = 1$). (ii) However, lowering the batch size to fit the model at $d = 5000$ in memory strongly deteriorates the performance compared to the baselines at $d = 1000$ (compare these results with those of Table 11). In other words, **the simple solution of increasing the embedding dimension, in addition to using vastly more parameters, comes at the cost of sacrificing critical hyperparameters such as batch size which deteriorates performance**.

Table 9: Full results for the FB15k-237 dataset.

| Model | FB15k-237 | | | | | | |
|---|---|---|---|---|---|---|---|
| | NLL↓ | MRR↑ | MR↓ | Hits@1↑ | Hits@3↑ | Hits@10↑ | Param |
| *d = 200* | | | | | | | |
| DISTMULT | 4.74±.08 | .304±.004 | 228±7.5 | .216±.003 | .331±.005 | .482±.004 | 3.0M |
| DISTMULT-MOS | 4.65±.01 | .306±.002 | 214±0.3 | .220±.002 | .336±.001 | .479±.003 | 3.3M |
| COMPLEX † | 4.74±.01 | .303±.001 | 220±2.3 | .216±.001 | .330±.002 | .482±.001 | 3.0M |
| COMPLEX-MOS † | 4.71±.01 | .301±.001 | 221±1.4 | .218±.001 | .329±.002 | .467±.001 | 3.3M |
| RESCAL | 4.79±.01 | .285±.001 | 246±2.3 | .210±.001 | .309±.001 | .432±.001 | 21.9M |
| RESCAL-MOS | 4.65±.01 | .318±.001 | 220±1.1 | .230±.002 | .348±.001 | .494±.001 | 22.2M |
| CONVE | **4.48±.01** | **.321±.001** | **176±2.6** | **.232±.000** | **.351±.001** | **.499±.001** | 5.1M |
| CONVE-MOS | 4.57±.01 | .311±.002 | 203±1.1 | .227±.002 | .339±.003 | .479±.002 | 5.4M |
| COMPGCN | 4.80±.07 | .300±.005 | 250±19.8 | .215±.004 | .328±.005 | .474±.008 | 1.6M |
| COMPGCN-MOS | 4.86±.03 | .310±.001 | 223±5.8 | .222±.001 | .339±.001 | .486±.001 | 2.0M |
| *d = 1000* | | | | | | | |
| DISTMULT | **4.56±.01** | **.331±.001** | 208±2.8 | **.241±.001** | **.362±.001** | **.514±.003** | 15.0M |
| DISTMULT-MOS | 4.72±.00 | .311±.001 | 231±2.3 | .221±.003 | .341±.000 | .492±.002 | 23.0M |
| COMPLEX † | 4.71±.45 | .317±.027 | 224±72.7 | .231±.019 | .347±.031 | .491±.047 | 15.0M |
| COMPLEX-MOS † | 4.64±.00 | .314±.001 | 226±3.1 | .225±.001 | .343±.001 | .497±.001 | 23.0M |
| RESCAL | 4.64±.00 | .307±.001 | 216±0.7 | .221±.001 | .335±.003 | .483±.002 | 488.5M |
| RESCAL-MOS | 4.63±.01 | .325±.002 | 259±5.4 | .236±.004 | .357±.002 | .505±.001 | 496.5M |
| CONVE | 4.65±.04 | .301±.001 | **185±4.3** | .212±.001 | .329±.001 | .485±.002 | 70.1M |
| CONVE-MOS | 4.69±.01 | .316±.001 | 222±1.5 | .227±.001 | .347±.001 | .497±.002 | 78.2M |

† Results for COMPLEX use halved embedding sizes $d = 100$ and $d = 500$.

Table 10: Full results for the Hetionet dataset.

| Model | Hetionet | | | | | | |
|---|---|---|---|---|---|---|---|
| | NLL↓ | MRR↑ | MR↓ | Hits@1↑ | Hits@3↑ | Hits@10↑ | Param |
| *d = 200* | | | | | | | |
| DISTMULT | 6.10±.00 | .250±.001 | 695±2.4 | .176±.001 | .274±.001 | .395±.001 | 9.0M |
| DISTMULT-MOS | 5.83±.00 | **.277±.002** | **488±1.3** | **.202±.003** | **.303±.002** | **.423±.002** | 9.4M |
| COMPLEX † | 6.10±.00 | .249±.001 | 698±2.8 | .174±.001 | .274±.001 | .395±.000 | 9.0M |
| COMPLEX-MOS † | 5.85±.00 | .269±.001 | 500±2.9 | .194±.001 | .294±.001 | .415±.001 | 9.4M |
| RESCAL | 6.13±.00 | .219±.001 | 607±0.4 | .153±.001 | .237±.001 | .351±.001 | 10.9M |
| RESCAL-MOS | 5.87±.01 | .274±.000 | 505±2.0 | .200±.000 | .300±.001 | .419±.001 | 11.3M |
| CONVE | 6.03±.00 | .252±.001 | 610±2.3 | .180±.001 | .275±.002 | .395±.002 | 11.1M |
| CONVE-MOS | 5.92±.00 | .263±.001 | 536±1.1 | .193±.002 | .284±.002 | .400±.001 | 11.4M |
| COMPGCN | 5.95±.01 | .260±.005 | 542±3.4 | .185±.005 | .285±.006 | .409±.006 | 4.7M |
| COMPGCN-MOS | 5.86±.08 | .266±.012 | 484±16.0 | .191±.010 | .292±.015 | .414±.018 | 5.0M |
| *d = 1000* | | | | | | | |
| DISTMULT | 6.04±.00 | .288±.000 | 633±1.9 | .217±.001 | .314±.000 | .429±.000 | 45.2M |
| DISTMULT-MOS | 5.76±.00 | .312±.001 | **491±1.5** | .233±.001 | **.344±.001** | **.467±.001** | 53.2M |
| COMPLEX † | 5.99±.00 | .292±.000 | 658±1.8 | .219±.001 | .320±.000 | .437±.000 | 45.2M |
| COMPLEX-MOS † | 5.78±.00 | .303±.000 | 504±2.1 | .223±.001 | .335±.001 | .459±.001 | 53.2M |
| RESCAL | 5.93±.00 | .243±.001 | 650±3.6 | .165±.001 | .270±.001 | .398±.001 | 93.1M |
| RESCAL-MOS | 5.87±.03 | .300±.009 | 542±4.9 | .223±.009 | .327±.010 | .454±.008 | 101.2M |
| CONVE | 6.06±.00 | .262±.001 | 635±4.7 | .187±.001 | .288±.001 | .411±.001 | 100.3M |
| CONVE-MOS | **5.71±.01** | **.313±.001** | 505±3.7 | **.237±.002** | .343±.001 | .462±.001 | 108.3M |

† Results for COMPLEX use halved embedding sizes $d = 100$ and $d = 500$.

Table 11: Full results for the ogbl-biokg dataset.

| Model | ogbl-biokg | | | | | | |
|---|---|---|---|---|---|---|---|
| | NLL↓ | MRR↑ | MR↓ | Hits@1↑ | Hits@3↑ | Hits@10↑ | Param |
| *d* = 200 | | | | | | | |
| DISTMULT | 4.83±.01 | .792±.001 | 5.60±0.1 | .713±.001 | .849±.001 | **.935±.001** | 18.8M |
| DISTMULT-MOS | 4.65±.00 | .792±.001 | 5.32±0.0 | .716±.002 | .844±.000 | .930±.000 | 19.1M |
| COMPLEX [†] | 4.83±.01 | .792±.001 | 5.56±0.0 | .713±.002 | .849±.001 | .935±.000 | 18.8M |
| COMPLEX-MOS [†] | **4.65±.00** | **.793±.001** | **5.26±0.0** | **.717±.002** | **.845±.001** | .931±.000 | 19.1M |
| RESCAL | 4.89±.00 | .763±.001 | 6.01±0.0 | .679±.001 | .818±.001 | .917±.000 | 22.8M |
| RESCAL-MOS | 4.70±.00 | .780±.001 | 5.47±0.0 | .699±.001 | .835±.001 | .928±.000 | 23.2M |
| CONVE | 4.94±.00 | .782±.001 | 5.57±0.0 | .701±.002 | .838±.001 | .928±.000 | 20.8M |
| CONVE-MOS | 4.77±.01 | .768±.002 | 5.77±0.1 | .683±.003 | .824±.002 | .923±.001 | 21.2M |
| COMPGCN [‡] | 5.11±.20 | .749±.019 | 7.00±0.7 | .665±.021 | .801±.019 | .905±.012 | 9.5M |
| COMPGCN-MOS [‡] | 4.96±.15 | .744±.022 | 6.93±0.9 | .659±.025 | .799±.022 | .904±.014 | 9.6M |
| *d* = 1000 | | | | | | | |
| DISTMULT | 4.89±.02 | .801±.002 | 5.96±0.1 | .726±.003 | .855±.002 | .936±.001 | 93.9M |
| DISTMULT-MOS | **4.34±.00** | **.837±.001** | **4.51±0.0** | **.772±.002** | **.884±.000** | **.951±.000** | 101.9M |
| COMPLEX [†] | 4.86±.00 | .806±.001 | 5.49±0.0 | .731±.002 | .860±.001 | .940±.000 | 93.9M |
| COMPLEX-MOS [†] | 4.39±.00 | .836±.001 | 4.50±0.0 | .770±.001 | .883±.000 | .951±.000 | 101.9M |
| RESCAL | 4.74±.00 | .799±.001 | 5.66±0.0 | .723±.001 | .851±.000 | .939±.000 | 195.8M |
| RESCAL-MOS | 4.42±.01 | .824±.004 | 4.87±0.1 | .755±.005 | .874±.002 | .947±.001 | 203.8M |
| CONVE | 4.93±.00 | .807±.000 | 5.25±0.0 | .731±.001 | .863±.000 | .941±.000 | 149.0M |
| CONVE-MOS | 4.43±.00 | .817±.001 | 5.17±0.1 | .744±.001 | .872±.000 | .946±.000 | 157.0M |

[†] Results for COMPLEX use halved embedding sizes *d* = 100 and *d* = 500.
[‡] Results for COMPGCN use *d* = 100 due to computational constraints.

Table 12: Full results for the openbiolink dataset.

| Model | openbiolink | | | | | | |
|---|---|---|---|---|---|---|---|
| | NLL↓ | MRR↑ | MR↓ | Hits@1↑ | Hits@3↑ | Hits@10↑ | Param |
| *d* = 200 | | | | | | | |
| DISTMULT | 5.14±.00 | .302±.002 | 1120±10.6 | .195±.004 | .342±.000 | .530±.000 | 36.2M |
| DISTMULT-MOS | **5.03±.01** | .314±.001 | 966±33.4 | .207±.001 | .351±.001 | .543±.000 | 36.5M |
| COMPLEX [†] | 5.13±.01 | .301±.001 | 1200±20.2 | .193±.002 | .340±.001 | .530±.001 | 36.2M |
| COMPLEX-MOS [†] | 5.06±.00 | .313±.002 | 909±18.5 | .206±.003 | .350±.002 | .539±.004 | 36.5M |
| RESCAL | 5.16±.00 | .303±.001 | 960±2.20 | .197±.002 | .339±.001 | .527±.001 | 38.4M |
| RESCAL-MOS | 5.04±.01 | **.323±.005** | 949±78.8 | **.217±.004** | **.361±.005** | **.547±.005** | 38.8M |
| CONVE | 5.28±.02 | .286±.000 | 794±17.6 | .181±.001 | .320±.002 | .509±.002 | 38.3M |
| CONVE-MOS | 5.10±.01 | .304±.002 | 967±47.9 | .200±.002 | .340±.003 | .528±.002 | 38.6M |
| COMPGCN [‡] | 5.43±.07 | .263±.007 | 573±27.0 | .162±.005 | .293±.009 | .480±.012 | 18.3M |
| COMPGCN-MOS [‡] | 5.35±.06 | .278±.001 | **515±35.9** | .174±.003 | .312±.004 | .495±.009 | 18.3M |
| *d* = 1000 | | | | | | | |
| DISTMULT | 5.17±.02 | .316±.004 | 1210±21.2 | .209±.004 | .357±.003 | .541±.004 | 181.0M |
| DISTMULT-MOS | 4.89±.01 | **.347±.000** | 799±32.1 | **.236±.002** | **.392±.002** | **.579±.002** | 189.0M |
| COMPLEX [†] | 5.12±.01 | .322±.001 | 1230±33.9 | .213±.001 | .364±.002 | .550±.001 | 181.0M |
| COMPLEX-MOS [†] | **4.87±.01** | .345±.001 | **725±18.7** | .235±.002 | .388±.001 | .575±.001 | 189.0M |
| RESCAL | 5.03±.00 | .328±.002 | 1330±26.6 | .222±.002 | .366±.001 | .552±.002 | 237.0M |
| RESCAL-MOS | 5.00±.01 | .330±.003 | 1210±27.8 | .220±.003 | .373±.003 | .559±.004 | 245.0M |
| CONVE | 5.24±.00 | .308±.001 | 1140±24.2 | .201±.001 | .348±.001 | .535±.001 | 236.2M |
| CONVE-MOS | 4.91±.01 | .336±.000 | 881±22.4 | .227±.001 | .378±.000 | .565±.001 | 244.2M |

[†] Results for COMPLEX use halved embedding sizes *d* = 100 and *d* = 500.
[‡] Results for COMPGCN use *d* = 50.

Table 13: Full results for the WN18RR dataset.

| Model | WN18RR | | | | | | |
|---|---|---|---|---|---|---|---|
| | NLL↓ | MRR↑ | MR↓ | Hits@1↑ | Hits@3↑ | Hits@10↑ | Param |
| $d = 200$ | | | | | | | |
| DISTMULT | 6.73±.02 | .421±.001 | 7220±2.5 | .397±.002 | .429±.001 | .469±.001 | 8.2M |
| DISTMULT-MOS | 9.08±.79 | .129±.154 | 8080±2000 | .107±.152 | .134±.159 | .171±.157 | 8.6M |
| COMPLEX [†] | 6.04±.04 | .436±.001 | 7640±347 | .413±.002 | .446±.001 | .479±.001 | 8.2M |
| COMPLEX-MOS [†] | 8.67±.38 | .143±.067 | 7590±563 | .112±.062 | .153±.071 | .202±.074 | 8.6M |
| RESCAL | 8.66±.54 | .215±.113 | 7640±2010 | .187±.116 | .225±.112 | .267±.104 | 9.1M |
| RESCAL-MOS | 8.91±.56 | .139±.137 | 7630±1730 | .112±.130 | .146±.145 | .188±.148 | 9.4M |
| CONVE | 7.31±.03 | .230±.002 | 4300±265 | .177±.003 | .250±.002 | .327±.004 | 10.3M |
| CONVE-MOS | 8.54±.41 | .116±.071 | 5610±1390 | .078±.056 | .127±.081 | .189±.102 | 10.6M |
| COMPGCN | 7.38±.13 | .411±.002 | 4550±167 | .386±.002 | .418±.002 | .459±.002 | 4.2M |
| COMPGCN-MOS | 9.04±.04 | .396±.001 | 5250±120 | .379±.002 | .402±.001 | .428±.003 | 4.6M |
| $d = 1000$ | | | | | | | |
| DISTMULT | 6.44±.01 | .438±.001 | 6380±70.9 | .405±.001 | .449±.002 | .507±.002 | 41.1M |
| DISTMULT-MOS | 6.02±.02 | .435±.002 | 5650±106 | .413±.002 | .443±.002 | .477±.002 | 49.2M |
| COMPLEX [†] | 5.67±.02 | .460±.001 | 6970±187 | .429±.001 | .475±.001 | .520±.001 | 41.1M |
| COMPLEX-MOS [†] | 5.93±.17 | .439±.003 | 5310±63.7 | .417±.001 | .446±.004 | .481±.008 | 49.2M |
| RESCAL | 8.14±.96 | .378±.020 | 5750±960 | .357±.018 | .387±.021 | .417±.023 | 63.1M |
| RESCAL-MOS | 6.79±1.58 | .421±.028 | 4820±175 | .394±.032 | .433±.024 | .469±.024 | 71.1M |
| CONVE | 5.55±.01 | .433±.001 | 3860±25.2 | .394±.001 | .448±.001 | .511±.002 | 96.2M |
| CONVE-MOS | 7.26±.15 | .416±.010 | 4400±62.3 | .391±.012 | .427±.009 | .462±.009 | 104.3M |

[†] Results for COMPLEX use halved embedding sizes $d = 100$ and $d = 500$.

Table 14: MRR and NLL on `ogbl-biokg` at $d = 1000$ with different numbers of softmaxes $K$.

| MODEL | $K$ | NLL ↓ | MRR ↑ | MODEL | $K$ | NLL ↓ | MRR ↑ |
|---|---|---|---|---|---|---|---|
| DISTMULT | 1 | 4.89±0.02 | .801±.002 | COMPLEX | 1 | 4.86±0.00 | .806±0.001 |
| DISTMULT-MOS | 1 | 4.42±0.01 | .821±.000 | COMPLEX-MOS | 1 | 4.46±0.01 | .820±0.001 |
| DISTMULT-MOS | 2 | 4.37±0.01 | .831±.001 | COMPLEX-MOS | 2 | 4.41±0.02 | .827±0.002 |
| DISTMULT-MOS | 4 | 4.34±0.01 | .837±.001 | COMPLEX-MOS | 4 | 4.39±0.00 | .836±0.001 |
| DISTMULT-MOS | 8 | 4.33±0.00 | .841±.001 | COMPLEX-MOS | 8 | 4.37±0.00 | .838±0.001 |

# G   COMPARISON OF KGE-MOS WITH ENSEMBLE MODELS

KGE-MOS can be reminiscent of ensemble models proposed in, e.g., Wang et al. (2017) for bilinear KGEs. However, the ensemble technique is still bottlenecked as it combines models linearly, and has a high parameter cost as it requires a separate model for each component. In contrast, KGE-MOS uses a single model with several output heads for a low parameter cost, and explicitly uses non-linear transformations to break the rank bottleneck.

Specifically, the ensemble model of Wang et al. (2017) calculates the final scores as

$$\sum_{k=1}^{K} w_k \mathbf{h}_{s,r}^k \mathbf{E}_k^\top$$

with $w_i$ combining a linear renormalization factor and an ensemble parameter. Compare this with the KGE-MOS scores

$$\sum_{k=1}^{K} \pi_k(\mathbf{h}_{s,r}) \operatorname{softmax}(f_k(\mathbf{h}_{s,r}) \mathbf{E}^\top)$$

there are two important differences to highlight:

**On breaking rank bottlenecks:** the ensemble technique is still linear with a rank bottleneck of $Kd$. Indeed, it can be rewritten as the factorization

$$[w_0 \mathbf{h}_{s,r}^0; \ldots; w_K \mathbf{h}_{s,r}^K][\mathbf{E}_0^\top; \ldots; \mathbf{E}_K^\top]$$

with $[;]$ a concatenation operation on the first axis. In contrast, KGE-MOS uses $K$ softmax layers *within* the mixture, which prevents this factorization and is necessary to break the bottlenecks. Notice also that the mixture weights in KGE-MOS depend on the query embedding, which is not the case for the ensemble model. Though this is not explicitly relevant to rank bottlenecks, it is another important difference between the two approaches.

**On parameter efficieny:** the ensemble model uses $k = 1, \ldots, K$ output embedding spaces $\mathbf{E}_k$, which costs $K$ times more parameters. In contrast, KGE-MOS uses projections $f_k$ of the query embedding for a low parameter cost. For example, with DISTMULT/COMPLEX, this costs only 4% more parameters in `openbiolink`, the largest dataset used in the experiments.

