# OpenReview forum: "Breaking Rank Bottlenecks in Knowledge Graph Embeddings"
_ICLR.cc/2026/Conference — Submitted to ICLR 2026_

### Official Review · Reviewer_rUSu · 2025-10-28

**Soundness:** 3
**Presentation:** 3
**Contribution:** 3
**Rating:** 6
**Confidence:** 3

**Summary:**

This paper investigates rank bottlenecks in knowledge graph embedding (KGE) models caused by the common low‑rank linear output layer. The authors theoretically show that such bottlenecks limit expressivity for ranking, sign, and distributional reconstruction tasks. They introduce KGE‑MoS, a mixture‑of‑softmaxes output layer that nonlinearly combines multiple softmax components to break low‑rank constraints at low parameter cost. Experiments on multiple large‑scale datasets demonstrate that KGE‑MoS consistently improves ranking accuracy and probabilistic fit compared with standard KGEs.

**Strengths:**

- The authors analyze the rank bottleneck problem in knowledge graph embedding (KGE) models and provide comprehensive theoretical constraints and inexpressibility proofs under three common tasks (DR, SR, and RR).
- A lightweight and general method, KGE‑MOS, is proposed by replacing the standard linear output layer with a mixture‑of‑softmaxes.
- The proposed method is evaluated on four datasets (FB15k‑237, Hetionet, ogbl‑biokg, and openbiolink) and achieves performance improvements on large knowledge graphs.

**Weaknesses:**

- The proposed method may not scale effectively to extremely large graphs (e.g., tens of millions of entities), where increasing embedding dimensions remains a common approach to enhance expressivity.
- The evaluation currently involves a limited set of relation‑rich datasets; it would be valuable to include additional experiments on large open KGs such as Wikidata.
- The performance degradation on smaller datasets indicates potential instability, which requires a clearer explanation.
- The training time increases due to multiple softmax computations.

**Questions:**

See the weaknesses

---

> ### Author Response · Authors · 2025-11-16
>
> We thank the reviewer for their useful feedback and for appreciating the comprehensiveness of our analysis, the generality and simplicity of our solution, and the quality of our experimental evaluations. Below we provide answers to weaknesses and questions.
>
> > **W1.** The proposed method may not scale effectively to extremely large graphs (e.g., tens of millions of entities), where increasing embedding dimensions remains a common approach to enhance expressivity.
>
> KGE-MoS would actually be parameter-efficient in graphs with tens of millions of entities. Given that GPU RAM memory is a hard limitation, the parameter cost of increasing the embedding dimension in such graphs would be prohibitive. Consider for example, Appendix F4.4 and Table 8, where we try to break bottlenecks in `ogbl-biokg` by simply increasing the dimension of `DistMult` to 5000\. To do so,
> * (i) not only we must use a very high number of params (400% more params, compared with MoS which only uses \~5% more params),
> * (ii) **increasing embedding size also forced us to decrease the batch size to fit in GPU RAM, which greatly deteriorates performance.**
>
> MoS specifically targets such large graphs and is a better solution to increase expressivity reliably without hurting training performance.
>
> > **W2.** The evaluation currently involves a limited set of relation‑rich datasets; it would be valuable to include additional experiments on large open KGs such as Wikidata.
>
> We will try to run experiments on `Wikidata5M` but it might take some time due to our limited computing budget. However, notice that while it has more entities, **`Wikidata5M` is much less densely connected than the datasets we already analyze:** the max $(s,r,\\cdot)$ out-degree in `Wikidata5M` is 1,352, which is much smaller than e.g. `ogbl-biokg` and its max out-degree 29,328. In fact, it is even smaller than `FB15K-237` (max out-degree 4,364).
>
> Thus, according to our theory in Section 4, we expect negative results in `Wikidata5M` as it would be a dataset easier to fit where rank bottlenecks might not matter. Still, showing this in the manuscript would be an interesting discussion highlighting that KG size *and* KG connectivity matter.
>
> > **W3.** The performance degradation on smaller datasets indicates potential instability, which requires a clearer explanation.
>
> We believe the degradation on small datasets does not reflect instability but simply overfitting. Appendix F.4, Tables 10-12 with **full results and standard deviations on large, dense graphs show minimal deviations across runs, hinting that the method is stable.** We think that KGE-MOS, by increasing the expressivity of the output layer, which is beneficial on large, dense graphs, can **overfit sparse, low-connectivity datasets** such as `FB15k-237` that provide too little signal per query. This behavior is consistent across the other small dataset (`WN18RR`, Appendix F.4 Table 13\) confirming that it is due to limited data rather than instability of the method. **We have clarifed this in our revision and hint to standard deviations in the Appendix.**

---

### Official Review · Reviewer_g8md · 2025-10-30

**Soundness:** 2
**Presentation:** 2
**Contribution:** 1
**Rating:** 0
**Confidence:** 5

**Summary:**

Argues that factorization-based KGE models produce a low-rank scoring matrix so that its representational power is limited by a rank bottleneck. Then transforms entity-relation embeddings uses multiple different non-linear transformations and subsequently ensembles; this increases the rank of the scoring matrix at low additional cost. Reports on a small experimental study using multiple KGE models.

I recommend to reject this paper because it is oblivious to key related work (W1), it partially does not study the actually relevant problems (W2), and it's experimental study is not convincing (W3).

**Strengths:**

Hard to say, as key related work is not represented.

**Weaknesses:**

W1. Not novel and key related work missing. The rank bottleneck has been studied in more detail, using tighter bounds, and applied to more models in [A]. It also proposes an ensemble approach. This paper is neither cited nor discussed.

W2. Does not use the right problems. The paper asks whether a KGE model can express every ranking. That's not relevant, however, if a KGE model can express every ranking, but only whether it can rank positives higher than negatives: the relative ranking of, say, two positives does not matter.

W3. Experimental results not convincing. Improvements of FB15k-237 are tiny and results (in terms of MRR) fall behind what has been reported for the baseline models in other papers. This reduces trust in the other datasets, which haven't been used that much in related work as far as I know. Datasets such as Wikidata5M, which have been used, are missing.

Wang et al., On Multi-Relational Link Prediction with Bilinear Models, AAAI, 2018

**Questions:**

-

---

> ### Author Response · Authors · 2025-11-16
>
> *(part 1/2)*
>
> We thank the reviewer for providing a useful and related paper which we missed. We will make sure to thoroughly discuss it in our manuscript. We would like to highlight why our results are novel compared to (Wang et al, 2018).
>
> **W1. [Novelty and missing related work]** After a careful read of (Wang et al, 2018), we find that the novelty of our contribution (i) needs to be further specified. On the other hand, our contributions (ii) and (iii) remain novel. Our experimental evaluation also specifically targets larger graphs than (Wang et al, 2018), which is where we argue rank bottlenecks matter. Going over our contributions:
>
> > re Contribution (i): We are the first to highlight the implications of rank bottlenecks on the three KGC prediction tasks in Figure 2 (Section 3).
>
> (Wang et al, 2018\) indeed studies rank bottlenecks in bilinear models for universal (“express any”) and specific (“express one correct”) ranking reconstructions. They also study, implicitly, specific sign reconstructions. However,
> * (i) where (Wang et al, 2018\) only targets specific bilinear embedding models, our results apply to any bilinear embedding model and neural network models such as CompGCN or transformer models,
> * (ii) we provide a bound for universal sign reconstruction (Theorem 3.3), which is not in (Wang et al, 2018),
> * (iii) we address the additional task of distributional reconstruction, highlighting its challenging implications in generative modeling, uncertainty quantification and complex query answering settings (Arakelyan et al, 2021; Loconte et al, 2023; Zhu et al, 2024). We have updated our claim to better reflect the existing work in rank bottlenecks for ranking reconstruction. Please see our revision.
>
> Arakelyan et al, 2021, Complex Query Answering with Neural Link Predictors
>
> Loconte et al, 2023, How to turn your knowledge graph embeddings into generative models
>
> Zhu et al, 2024, Conformalized Answer Set Prediction for Knowledge Graph Embedding
>
> > re Contribution (ii): In Section 4, we provide sufficient dimensionality conditions to perfectly break bottlenecks in some tasks and show that the task difficulty depends on the graph connectivity.
>
> Notice that our sufficiency condition for representing a specific knowledge graph is novel. (Wang et al, 2018\) also discuss conditions for expressing a *specific* Knowledge Graph instance (our Section 4\) in ranking and sign reconstructions. Yet, **their condition for specific reconstruction depends on the rounding rank of a matrix, which is NP-hard and cannot be determined in practice.** In Section 4, **our Corollary 4.2 gives an upper-bound on this rounding rank that depends on the graph connectivity and is easily computed in practice.** It is proved with an interpolation solution (Theorem 4.1 and ordering discussion in Appendix B). This is what allows us to derive the novel bounds for several datasets in Table 1. We clarified this difference in Section 4.
>
> > re Contribution (iii): We introduce KGE-MOS, a simple output layer that breaks rank bottlenecks and improves performance in large-scale benchmarks at a low parameter cost (Section 5).
>
> KGE-MoS is still the first solution that breaks the linearity of the output layer in KGEs and outputs high-rank scoring matrix.
> The ensemble technique in (Wang et al, 2018) (i) is still bottlenecked as it combines models linearly, (ii) has a high parameter cost as it requires a separate model for each component. In contrast, KGE-MOS uses a single model with several output heads for a low parameter cost, and explicitly uses non-linear transformations to break the rank bottleneck.
>
> Specifically, the ensemble model described in (Wang et al, 2018) mixes KGEs as
> $$\sum\limits_{k=1}^K w_k h^k_{s,r} E_k^\top$$
> with $h^k_{s,r} \in \mathbb{R}^d$, $E_k \in \mathbb{R}^{N\times d}$ and $w_i$ combining a linear renormalization factor and an ensemble parameter. Comparing this with the KGE-MoS scores
> $$\sum\limits_{k=1}^K \pi_k(h_{s,r}) \operatorname{softmax}(f_k(h_{s,r}) E^\top)$$
> we can highlight:
>
> 1. **On breaking rank bottlenecks:** the ensemble technique is still linear with a rank bottleneck of $Kd$. Indeed, it can be rewritten as the factorization
> $$[w_0 h_{s,r}^0;\dots;w_K h_{s,r}^K] [E_0^\top; \dots; E_K^\top]$$
>  with $[;]$ a concatenation operation on the first axis. MoS uses $K$ softmax layers *within* the mixture, which prevents this factorization and is necessary to break bottlenecks. Notice also that the mixture weights in KGE-MoS depend on the query embedding, which is not the case for the ensemble model.
> 2. **On parameter efficiency:** the ensemble model uses $k=1,\dots,K$ output embedding spaces $\mathbf{E_k}$, which costs $K$ times more parameters. In contrast, KGE-MoS uses projections of the query embedding for a low parameter cost. For example, with DistMult/Complex, this costs only 4% more parameters in `openbiolink`, the largest dataset.
>
> We added this comparison in our related work and in Appendix G.

---

> > ### Author Response · Authors · 2025-11-16
> >
> > *(part 2/2)*
> >
> >
> > **W2. [Universality vs specific expressivity]** We want to raise awareness to **our Section 4, which tackled the precise problem raised by the reviewer.** "*An important question remains: is there a bottlenecked model which, while unable to represent all possible results, can accurately represent the ground truths relevant to the knowledge graph? We address this by (... rest of Section 4\)*"
> >
> > **W3. [Experimental results]** Please note that the lack of improvements on FB15K-237 agree with our claims. We specifically raise the issue for large, dense graphs, and our results show **gains are strongest exactly where theory predicts they matter.** As we highlight in our Results section: **bottlenecks are not an issue on small-scale academic datasets** like FB15K-237. “*In contrast, on the smallest dataset FB15k-237, KGE-MOS does not improve performance (...), hinting that rank bottlenecks are not critical in small-scale datasets.*”
> >
> > We also want to highlight that **our large and dense datasets are also commonly used**. For example, `ogbl-biokg` comes from the "Open Graph Benchmark" NeurIPS 2020 paper, which has \~4k citations.
> >
> > We will try to run experiments on `Wikidata5M,` but it might take some time due to our limited computing budget. However, notice that while it has more entities, **`Wikidata5M` is much less densely connected than the large datasets we already analyze**: the max $(s,r,\\cdot)$ out-degree in `Wikidata5M` is 1,352, which is much smaller than e.g. `ogbl-biokg` and its max out-degree 29,328. In fact, it is even smaller than `FB15K-237` (max out-degree 4,364).  According to our theory in Section 4, this makes `Wikidata5M` a dataset much easier to fit where rank bottlenecks might not matter.

---

> ### Comment · Reviewer_g8md · 2025-11-17
> **Thoughts on response**
>
> On W1. I agree that the arguments of Wang et al. (2018) are w.r.t. to bilinear embedding models. They are, however, closely related to what is done in this submission in terms of approach and methodology. From a theoretical perspective, it's not clear what this submission adds: some proofs directly follow from prior work, results are not put into perspective and compared to Wang et al., and its not clear to what extent additional work on bounding sign ranks exists and is relevant. Some of the results are obvious (e.g., Prop 3.1). Also, the fact that the "task difficulty depends on the graph connectivity" is first shown in this paper is misleading: it is instead immediately clear given the relationship to sign rank. In short: A thorough discussion is needed and lacking.
>
> On W2. Here the authors state that they look at ground-truth representation in Sec 4, which I agree to. However, they do not consider any actual KGE model, but instead assume that "H" can be arbitrarily chosen. This result now follow from a sign rank-decomposition  (again, akin to the approach of Wang et el.) and any bounds on sign ranks apply directly. The key question---whether any known KGE model can provably obtain such an representation---is not answered.
>
> On W3. KGE-MoS is, as the paper clearly states, is not a new approach in general, but it's first applied to KGE models here. Nevertheless, (i) it is not analyzed (along the lines of the prior theorems) and (ii) it's performance is not adequately put into context. Even if we discounted FB15k-237 as suggested by the authors, SOTA results are still not discussed (e.g., Loconte et al.'s results on OGBL-BIOKG).
>
> Overall, I feel that more due diligence is needed to this work. This is the main rationale for my recommendation to reject the paper.

---

> > ### Author Response · Authors · 2025-12-01
> >
> > We thank the reviewer for engaging in the conversation and going through our arguments.
> >
> > On W1 \[**Novelty and discussions w.r.t. Wang et al (2018)**\] We have reorganized our theoretical results to help contextualise the difference to Wang et al (2018). We now explicitly highlight which results are for *universality*, and which are for *realisability* (i.e., expressing a particular graph).  – Note that Wang et al also tackle both, and it is important to compare these results.
> > Then, when introducing our general result for realisability in Ranking Reconstruction:
> >
> > * We explain that while **Wang et al (2018) derive a *sufficient* condition** for ranking realisability in *some KGEs* (e.g., RESCAL, ComplEx), **we derive a *necessary* condition** for ranking realisability in *all KGEs, including those parameterised by expressive neural networks*.
> > * We concretely relate these bounds in the new Appendix E.2. The sufficient bound on rank dimension for RESCAL/ComplEx from Wang et al (2018) is up to $2 |\mathcal{R}|$ larger than our necessary bound, and so they act respectively as lower and upper bounds: The necessary AND sufficient dimension for RESCAL/ComplEx is somewhere between our result and that of Wang et al.
> >
> > The rest of the results (distribution/sign reconstruction) are still under a different scope.
> >
> > Notice that this “sufficient vs necessary” distinction also highlights our angle compared to these previous works. **While some topics may be close, our message is very different**. Previous works (Wang et al, 2018 or specific KGE papers of ComplEx, Tucker, …) usually argue that "KGE X is expressive enough with the right bounds". In contrast, we question the practicality of these bounds. We give **necessary conditions for all KGEs and highlight that these are impractical to meet in larger/denser graphs** and limit all KGEs. We argue for more exploration in non-linear and high-rank KGEs. **We believe this shift in position is also part of the significance of our work.** Still, in the latest revision, we discuss Wang et al (2018) in five locations: introduction, sec 3 (rankings), related work, appendix E2 (relating concretely the bounds) and G (difference with ensemble technique).
> >
> >
> > On W2 \[**Sufficiency condition in the context of existing KGEs**\] We  showed that the sufficiency condition grows for larger and denser graphs. *Even if* there are linear KGEs that can represent this solution, the bound becomes too large to be practical as graphs grow. Still, we agree that addressing whether known KGEs can express this representation is interesting. We have added the following paragraph:
> >
> > > (...) With large enough encoding layers, neural KGEs should be able to express this factorization by virtue of neural networks being universal approximators (the $h_{s,r}$ and $e_o$ representations are not rigidly tied, which allows to represent the proper embeddings for the factorization).
> > However, bilinear KGEs are more limited. The best known sufficient bound for RESCAL or Complex, shown by \[Wang et al, 2018\], is up to $2|\mathcal{R}|$ times larger than the one of $\operatorname{srank}(Y^\pm)$ (see Appendix E2). It is not clear where the necessary *and* sufficient bound lies between these two. Distmult, on the other hand, cannot realise the solution at all due to its symmetric score function. (...)
> >
> > On W3 \[**Analysis and comparisons of KGE-MoS**\] For the analysis of MoS: please note Appendix F.4.2 and Table 6\. We analyse KGE-MoS in the context of high-rank scores by showing it produces high-rank distributions.
> >
> > Concerning the comparisons with SOTA: **The goal of our paper is not to get the best results in each benchmark, nor is it part of its claims.** We do not position our output layer against other advancements in loss functions, fine-tuning strategies, etc. Instead, our empirical results show that KGE-MoS is easily applicable to many models, that it is useful in low and high dimensional regimes, and **that rank bottlenecks are an issue where the theory expects them to matter**.
> >
> > Q1 \[**Usefulness of universality results**\] We agree with the reviewer that in practice, we are only interested in the specific subset of feasible assignments that correspond to the ground truth, not the entire combinatorial space $2^{|\mathcal{E}|}$. This is why we focus on *realisability* in much of the paper. However, we believe the *universality* theorems are also important for two reasons:
> >
> > 1. **Contextualisation of previous work**: many recent KGEs approaches use universal guarantees as a motivation ("if it can represent any graph, it can represent the ground truth graph"). We give a lower bound on these to show that it requires at least $d=|\mathcal{E}|$, which is impractical for large KGs.
> > 2. **'Unknown ground truth' risk**: while we only care about the ground truth, we do not know it a priori (KGs are assumed incomplete in KGC). Without universality, we can't really guarantee that the true configuration is attainable.

---

> ### Comment · Reviewer_g8md · 2025-11-17
> **Afterthought**
>
> I also wonder about the usefulness of Th. 3.3; this is more of a question. Why do we care about this result? In a KG with K relations and E entities, we are interested in representing the K*E "true" assignments (i.e., not in representing all of the 2^E possible assignments).

---

### Official Review · Reviewer_EZ1a · 2025-10-31

**Soundness:** 3
**Presentation:** 3
**Contribution:** 2
**Rating:** 2
**Confidence:** 3

**Summary:**

A drop-in output layer for select Knowledge Graph Embedding models to increase model expressivity and thus predictive power by addressing "rank bottlenecks". The output layer consists in a mixture of softmaxes.

**Strengths:**

- Problem novelty: rank bottleneck problem not studied yet in KGE literature, to the best of my knowledge.
- Paper is well written, and it includes comprehensive material.
- Contribution is an original adoption of methods from language modelling literature.
- Evaluation: good mixture of benchmark datasets.

**Weaknesses:**

- The rank bottleneck problem could use a more in-depth introduction, to broaden up the audience.
- Contribution limited to adopting a MoS layer to existing KGE architectures.
- KGE-MOS does not support translation-based KGE methods (e.g. RotatE).
- Evaluation: limited impact of \*-MoS on predictive power. results at par with baselines.
- Experimental results presented in the paper does not justify the adoption of KGE-MOS in practice due to computational overhead (e.g. 2.75 slower to train)

**Questions:**

- Figure 1: Why adding a relation type as target object? Could you please elaborate?

---

> ### Author Response · Authors · 2025-11-16
>
> We thank the reviewer for their feedback and for appreciating our novelty, the writing of the paper and the breadth of our evaluation. We also thank the reviewer for noticing a typo in our first figure. Following are some comments on questions and feedback points.
>
> > W1. The rank bottleneck problem could use a more in-depth introduction, to broaden up the audience.
>
> This is a valuable suggestion to improve the accessibility of the paper. **We have added a small introductory section on rank bottlenecks before talking about KGC, please see our revision.**
>
> > W2.  Contribution limited to adopting a MoS layer to existing KGE architectures.
>
> We clarify that, while KGE-MOS adopts the general idea of Mixture-of-Softmaxes in the experiments, our contribution is not merely a direct application as it also provides theoretical and empirical aspects. We also:
>
> 1. We prove expressivity results of all bilinear models, as well as all newer neural network models. We also give the first formal analysis of the issue in the adjacent objectives of sign and distributional reconstructions.
> 2. We give a theoretical sufficient bound relating embedding dimension, ranking/sign reconstruction, and KG connectivity (Theorem 4). This result is independent of MoS and provides new insight into KGE expressivity.
> 3. We give empirical evidence that rank bottlenecks manifest differently in KGEs than in LM, due to entity set sizes and KG sparsity patterns, and show in three densely connected datasets that they matter and are solvable.
>
> So, **the MoS is only *one* contribution; the other core contributions concern theoretical analysis and empirical characterisation that are novel for the KGE domain.**
>
> > W3. KGE-MOS does not support translation-based KGE methods (e.g. RotatE).
>
> This is correct. We highlight this in our Related Work section. Our focus is the class of KGEs that compute object scores using vector-matrix multiplication because it includes the majority of KGE models used in recent benchmarks. We agree that extending the analysis and developing analogous “high-rank” output layers for translation-based models is an interesting direction for future work as we wrote in the Related Work and Conclusions.
>
> > W4. Evaluation: limited impact of \*-MoS on predictive power. results at par with baselines.
>
> We would like to clarify points that may not have been highlighted clearly in the paper:
>
> * On large, high-connectivity datasets, \-MoS shows consistent and statistically significant gains. All larger scale benchmarks (`Hetionet`, `ogbl-biokg`, `openbiolink`) show significant MRR and NLL improvements (p \< 0.05).
> * In other words, **gains are strongest exactly where theory predicts they matter.** The fact that `FB15K-237` (low connectivity) shows no gains is expected.
>
> We have added comments in the Results section to highlight this.
>
> > W5. Experimental results presented in the paper does not justify the adoption of KGE-MOS in practice due to computational overhead (e.g. 2.75 slower to train)
>
> We appreciate this point and also mention it in our paper. KGE-MoS presents a trade-off between parameter efficiency and inference overhead. In predictive performance, \-MoS at $d=200$ $\\approx$ baselines at $d=1000$; whereas \-MoS only uses \~20% of the parameter cost. **The inference slowdown is the cost of breaking a fundamental bottleneck without multiplying by 5 the parameter usage, which is not always possible when GPU RAM is a hard bottleneck.** Notice, for example, our results in Table 8, Appendix F4.4 where we instead increase the dimension of `DistMult` to 5000 to break bottlenecks by increasing embedding dimension. This (i) requires a prohibitively high number of params and (ii) a lower batch size, which greatly deteriorates performance. In such scenarios, \-MoS is the only available solution to increase expressivity reliably.
>
> > Q1. Figure 1: Why adding a relation type as target object? Could you please elaborate?
>
> We thank the reviewer for highlighting this, which was a typo from our side. The target object should have been `inflammation`. We have revised the Figure.

---

### Official Review · Reviewer_Wgvc · 2025-11-01

**Soundness:** 3
**Presentation:** 4
**Contribution:** 3
**Rating:** 6
**Confidence:** 4

**Summary:**

The paper looks into a limitation in tensor-multiplicative scoring functions used in KGE models, which the authors describe as a kind of "rank bottleneck" that prevents models from properly capturing complex relationships, especially in large graphs with high connectivity. To address this, they introduce KGE-MOS, which modifies the output layer by combining multiple softmax outputs. The idea is that this mixture can produce a more expressive scoring function and help models rank entities more accurately, particularly on large-scale knowledge graphs.

**Strengths:**

I think the paper’s main strength is its theoretical analysis of the bottleneck problem. The authors do a solid job of identifying and characterizing this limitation and even relate it to graph connectivity, which is interesting. The experiments are also convincing overall that KGE-MOS seems to improve results in the right settings (i.e., large graphs) without blowing up the parameter count. It’s a clear and well-motivated piece of work.

**Weaknesses:**

I do have a few concerns about the experimental part.

W1. The baselines are reasonable (DISTMULT, ConvE, etc.), but they’re a bit outdated. There are more recent KGE architectures — some Transformer- or GNN-based — that also use similar scoring layers. It would strengthen the argument a lot if the authors could show that their method helps even those stronger baselines, not just the classic ones.

W2. The ablation on the number of mixtures, $K$, is only done on DISTMULT and on a single dataset (ogbl-biokg). That’s informative, but it’s hard to know how general the trend is. Running at least one more model or dataset would make the case much stronger.

**Questions:**

Q1. The approach reminds me a bit of ensemble-based KGE methods (e.g., [1]). It might be worth clarifying how KGE-MOS is different from those, since both seem to combine multiple outputs to improve expressivity.

Q2. While the paper has a nice theoretical discussion of the rank bottleneck itself, the expressivity of the proposed fix is mostly justified through intuition and experiments. Have the authors considered analyzing the expressivity of KGE-MOS more formally? It would help complete the theoretical story.

---

> ### Author Response · Authors · 2025-11-16
>
> We thank the reviewer for their thorough feedback and for their suggestions for improvements. Furthermore, we are happy that the reviewer appreciates our theoretical motivations and finds our analysis rigorous and well-structured, and that our results are positive in the large-scale settings. Please find the answers to questions below. In particular, we clarified the difference with ensemble-based models and we are working on more ablation on the number of mixtures $K$ as suggested.
>
> **W1. [Transformer- or GNN- baselines]**  In Table 1 we show improvements on a GNN-based model (`CompGCN`), confirming that our theoretical results also apply to neural methods. We would like to test more of these complex models, but they are hard to apply on large-scale datasets because of their memory requirements (which is why we could only run `CompGCN` at $d=200$). What these models gain from their flexible score functions (they can incorporate information from a language model or be applied in inductive settings), they lose in scalability, making it hard to fit in our extensive study.
>
> **W2. [Mixture ablation]** We are currently running an extended ablation with DistMult and ComplEx on `ogbl-biokg` and `openbiolink`, our two datasets with the highest connectivity. Following are early results of ComplEx on `ogbl-biokg` that confirm the trend with DistMult.
> | Model         | NLL (↓) | MRR (↑) |
> |---------------|---------|---------|
> | Complex K=1   | 4.46    | 0.820   |
> | Complex K=2   | 4.41    | 0.827   |
> | Complex K=4   | 4.39    | 0.836   |
> | Complex K=8   | 4.37    | 0.838   |
> We hope to soon confirm these results with `openbiolink` and will update the revision with all results.
>
> **[Q1. difference with ensembles]** We thank the reviewer for pointing out the use of ensemble models in KGEs, which we were not aware of. The reference \[1\] is missing in the review, but we guess that it would be similar to (Wang et al, 2018), also referred by reviewer g8md. We show next that, unlike KGE-MoS, the ensemble technique (i) is still bottlenecked as it combines models linearly, (ii) has a high parameter cost as it requires a separate model for each component. In contrast, KGE-MOS uses a single model with several output heads for a low parameter cost, and explicitly uses non-linear transformations to break the rank bottleneck.
>
> Specifically, the ensemble model described in (Wang et al, 2018\) mixes KGEs as
> $$\sum\limits_{k=1}^K w_k h^k_{s,r} E_k^\top$$
> with $h^k_{s,r} \in \mathbb{R}^d$, $E_k \in \mathbb{R}^{N\times d}$ and $w_i$ combining a linear renormalization factor and an ensemble parameter. Comparing this with the KGE-MoS scores
> $$\sum\limits_{k=1}^K \pi_k(h_{s,r}) \operatorname{softmax}(f_k(h_{s,r}) E^\top)$$
> we can highlight:
>
> 1. **On breaking rank bottlenecks:** the ensemble technique is still linear with a rank bottleneck of $Kd$. Indeed, it can be rewritten as the factorization
> $$[w_0 h_{s,r}^0;\dots;w_K h_{s,r}^K] [E_0^\top; \dots; E_K^\top]$$
>  with $[;]$ a concatenation operation on the first axis. MoS uses $K$ softmax layers *within* the mixture, which prevents this factorization and is necessary to break bottlenecks. Notice also that the mixture weights in KGE-MoS depend on the query embedding, which is not the case for the ensemble model.
> 2. **On parameter efficiency:** the ensemble model uses $k=1,\dots,K$ output embedding spaces $\mathbf{E_k}$, which costs $K$ times more parameters. In contrast, KGE-MoS uses projections of the query embedding for a low parameter cost. For example, with DistMult/Complex, this costs only 4% more parameters in `openbiolink`, the largest dataset.
>
> We added this comparison in our related work and in Appendix G.
>
> Wang et al., On Multi-Relational Link Prediction with Bilinear Models, AAAI, 2018
>
> **Q2. [Expressivity of MoS]** We unfortunately haven't found a way to derive a closed-form solution for the output scores' rank in a Mixture-of-Softmaxes layer. Still, we attempt to prove "by example" in Appendix F.4.2 that the layer outputs high-rank solutions. On small datasets (`FB15K-237`) we can calculate the full score matrix and measure its rank; Table 6 shows that increasing K quickly increases this empirical rank, with K=4 at d=1000 resulting in a full-rank matrix (14,541) on this dataset.

---

> ### Comment · Reviewer_Wgvc · 2025-11-28
>
> Thank the authors for their responses. The additional results on ComplEx satisfy my concern regarding the generality of the work. Please ensure these are included in the final version. Since my main concerns about the ablation and the theoretical differences from ensembles are addressed, I'll raise my score.

---

### Author Response · Authors · 2025-12-03
**Global Response**

We thank the Area Chair for taking over our submission under challenging circumstances and appreciate the effort invested in carefully evaluating our work. Below, we provide a concise summary of how the revised manuscript addresses the core concerns raised in the reviews.

Firstly, we appreciate that reviewers $\text{\textcolor{red}{Wgvc}}$, $\text{\textcolor{purple}{EZ1a}}$ and $\text{\textcolor{green}{rUSu}}$ agree that our paper is clear, theoretically well-motivated, novel and has a comprehensive evaluation. Reviewers $\text{\textcolor{red}{Wgvc}}$ and $\text{\textcolor{green}{rUSu}}$ believe that our results show improvements in the right setting (large graphs) without blowing up the parameter count.

Reviewers $\text{\textcolor{red}{Wgvc}}$ and $\text{\textcolor{blue}{g8md}}$ asked about the differences between our \-MoS solution and ensemble-based techniques \[Wang et al, 2018\]. In the new Appendix G, we show that ensembles linearly combine several models, meaning that they do not break rank bottlenecks (while having a high parameter cost). In contrast, \-MoS is a non-linear augmentation of a single model (breaks bottlenecks \+ parameter efficient). Following this resolution, as well as a further ablation on the number of mixtures, *reviewer $\text{\textcolor{red}{Wgvc}}$ had intended to raise their score*.

Reviewer $\text{\textcolor{green}{rUSu}}$ questioned the scalability of \-MoS compared to increasing the embedding size $d$. In Appendix F4.4 and Table 8, we find that \-MoS scales better in (i) parameter cost, (ii) inference memory usage, and (iii) performance. Furthermore, as suggested by Reviewer $\text{\textcolor{purple}{EZ1a}}$, we have added a short background section on rank bottlenecks to broaden up the audience.

The main criticisms came from reviewers $\text{\textcolor{blue}{g8md}}$ and $\text{\textcolor{purple}{EZ1a}}$, the latter still listing many strengths to our work. Unfortunately, because of the changes in ICLR’s reviewing process, we could not finish our discussion, even though we have clear answers for their concerns.

The main concern of Reviewer $\text{\textcolor{blue}{g8md}}$ is the novelty of our results compared to \[Wang et al, 2018\], a reference we missed. We now thoroughly discuss it in the manuscript as it is indeed relevant to ours. However, we provided evidence that our theoretical and empirical results are different from \[Wang et al, 2018\]’s, which

1. Only studies one task (ranking reconstruction) while we study three (ranking, sign and distribution reconstruction).
2. Gives *sufficient* conditions for *five bilinear KGEs*, while we give *necessary* conditions in *all KGEs, including those parameterised by expressive neural networks* (sec 3). We concretely relate Wang’s bounds to ours in the new Appendix E2: Their smallest sufficient bound (for ComplEx/RESCAL) is up to $2 |\\mathcal{R}|$ times larger than our necessary bound, and so they act respectively as lower and upper bounds.
3. Use an ensemble solution which does not break rank bottlenecks, unlike our MoS solution (as discussed above and approved by reviewer $\text{\textcolor{red}{Wgvc}}$). Additionally, their experimental study does not focus on large and dense datasets.

The second concern of reviewer $\text{\textcolor{blue}{g8md}}$ was that we did not discuss a certain type of expressivity. We would like to clarify that *section 4 of our paper covered exactly that type of expressivity*. To avoid future confusion, we highlighted this expressivity under a specific term (*realisability*) in our revision, and emphasized it in each relevant result.

The final concern of reviewer $\text{\textcolor{blue}{g8md}}$, shared by reviewer $\text{\textcolor{purple}{EZ1a}}$, is the lack of performance improvement in FB15K-237 (our dataset with low node degree). While this is accurate, it is not part of our claims and agrees with our theory. Our improvements are strong and statistically significant exactly where theory predicts they matter: on large and dense graphs. We clarified this in the paper.

\[Wang et al, On Multi-Relational Link Prediction with Bilinear Models, AAAI, 2018\]

---

### Meta-Review · Area_Chair_VaDy · 2026-01-04

**Summary:**

The paper ivestigates the  limitation caused by scoring functions relying on simple vector matrix multiplication used in KGE models, which builts a kind of bottleneckt that restricts expressivity of the model. To address this, they  suggest to modify the output layer by combining multiple softmax outputs. The experiments show that this mixture can produce a more expressive scoring function and help models rank entities more accurately, particularly on large-scale knowledge graphs.

**Reviewer Concerns:**

Reviewers main concerns were:
1. baselines are outdated and paper should include more recent KGE architectures, e.g. neural netwroks based ones. Athuros pointed out that they already evaluate on neural network based model and that most neural network models can not be applied to larger KGs.
2. limmited models and datasets in ablation study on k. Authors added new experiments.
3. missing comparistion to existing work that studied rank bottlenecks and ensemble-style fixes ( Wang et al., 2018). The authors clarified differences to Wang et al. (2018), reframed claims, distinguished necessary vs. sufficient bounds, and added an elaborated discussion to the appendix. This did not fully convince all reviewrs.  Concerns were raised that parts of the theory focus on universality or arbitrary rankings rather than the practically relevant task of ranking positives above negatives, and that some results follow directly from known sign-rank arguments.

The rebuttal solved some but not all concerns. Concerns not resolved regard novelty and theoretical depth relative to prior work. The paper imrproved but is still boarderline.

**Reviewer Scores:**

Reviewer Wgvc pointed out that he will raise his score.
Reviewer g8md  who is an expert in the field still formulated doubts about novelty, so I expect he would not (or only little) have risen his score, still voting for rejection. Therefore, the paper overall remains boarderline.

---

### Decision · Program_Chairs · 2026-01-26

Reject